# Neural activity induces strongly coupled electro-chemo-mechanical interactions and fluid flow in astrocyte networks and extracellular space—A computational study

**Marte J. Sætra** *, Ada J. Ellingsrud, Marie E. Rognes

Department of Numerical Analysis and Scientific Computing, Simula Research Laboratory, Oslo, Norway

* martejulie@simula.no

**Data Availability Statement:** The code described in the paper is freely available online at https://

## Abstract

The complex interplay between chemical, electrical, and mechanical factors is fundamental to the function and homeostasis of the brain, but the effect of electrochemical gradients on brain interstitial fluid flow, solute transport, and clearance remains poorly quantified. Here, via in-silico experiments based on biophysical modeling, we estimate water movement across astrocyte cell membranes, within astrocyte networks, and within the extracellular space (ECS) induced by neuronal activity, and quantify the relative role of different forces (osmotic, hydrostatic, and electrical) on transport and fluid flow under such conditions. We find that neuronal activity alone may induce intracellular fluid velocities in astrocyte networks of up to 14μm/min, and fluid velocities in the ECS of similar magnitude. These velocities are dominated by an osmotic contribution in the intracellular compartment; without it, the estimated fluid velocities drop by a factor of ×34–45. Further, the compartmental fluid flow has a pronounced effect on transport: advection accelerates ionic transport within astrocytic networks by a factor of ×1–5 compared to diffusion alone.

## Author summary

Over the last decades, the neuroscience community has paid increased attention to the astrocytes—star-shaped brain cells providing structural and functional support for neurons. Astrocyte networks are likely to be a crucial pathway for fluid flow through brain tissue, which is essential for the brain's volume homeostasis and waste clearance. However, numerous questions related to the role of osmotic pressures and astrocytic membrane properties remain unanswered. There are also substantial gaps in our understanding of the driving forces underlying fluid flow through brain tissue. Answering these questions requires a better understanding of the interplay between electrical, chemical, and mechanical forces in brain tissue. Due to the complex nature of this interplay and experimental limitations, computational modeling can be a critical tool. Here, we present a high fidelity computational model of an astrocyte network and the extracellular space. The model predicts the evolution in time and distribution in space of intra- and extracellular volumes,

github.com/martejulie/fluid-flow-in-astrocyte-networks.

**Funding:** This project has received funding from the European Research Council (ERC) under the European Union's Horizon 2020 research and innovation programme under grant agreement #714892 (received by MER) and the Research Council of Norway (RCN) via FRIPRO grant agreement #324239 (EMIx, received by MER). The funders had no role in study design, data collection and analysis, decision to publish, or preparation of the manuscript.

**Competing interests:** The authors have declared that no competing interests exist.

ion concentrations, electrical potentials, and hydrostatic pressures following neural activity. Our findings show that neural activity induces strongly coupled chemical-mechanical-electrical interactions in the tissue and suggest that chemical gradients inside astrocyte syncytia strengthen fluid flow at the microscale.

# 1 Introduction

The complex interplay between chemical, electrical, and mechanical factors is fundamental to central nervous system physiology [1–3] and pathologies such as edema and stroke [4], vascular dementia, and neurodegenerative disease [5, 6], as well as in cortical tissue engineering [7]. Movement of ions between neuronal, glial, and extracellular spaces underpin membrane depolarization and thus the electrical activity of excitable cells [8–10]. On the other hand, chemical gradients induce osmotic pressures forcing water across the semi-permeable cellular membranes [11–13], challenging their volume homeostasis and inducing hydrostatic pressure gradients within each compartment. Simultaneously, the pulsating mechanical forces of the cardiovascular system act on this environment at the neuro-vascular and glio-vascular interfaces.

The renewed interest in brain clearance pathways over the last decade, supported by breakthroughs in imaging [14], experimental discoveries [15, 16], and the potential of computational modeling, has brought new relevance and new perspectives to this intriguing interplay. Key open questions relate to the role of astrocytes in brain signaling, volume homeostasis, and clearance in general [3, 16], and the role of osmotic pressures and astrocytic membrane properties on brain solute transport in particular. Specifically, there are substantial gaps in our understanding of (i) the driving forces underlying interstitial fluid movement and (ii) biophysical mechanisms for how astrocyte membrane features such as AQP4- and KIR-channels or NKCC1 co-transporters contribute to perivascular or interstitial transport [3, 16–20].

These questions seem intrinsically related to interactions between mechanical and chemical forces, and molecular diffusion juxtaposed with electrical and advective drift, but have largely not been addressed as such. The focus in computational neuroscience has conventionally been on electrophysiology alone [21], predominantly neglecting intracellular or extracellular ionic gradients [22]. In turn, even models at extreme morphological detail of intracellular and extracellular diffusion and reactions tend to ignore electrical drift and fluid mechanics [23, 24]. Moreover, while there has been a surge of computational fluid dynamics studies of the magnitude and mechanisms of cerebrospinal fluid flow, perivascular fluid flow [25, 26], and interstitial fluid flow [27–29] and their effect on brain solute transport [30–33], almost all ignore electro-chemical and osmotic effects.

In Halnes et al. [34], the authors introduce an electrodiffusive framework for modeling ion concentrations in astrocytes and extracellular space along one spatial dimension while ignoring mechanical aspects such as cellular swelling and compartmental fluid dynamics. By modeling ionic concentrations in intra- and extracellular compartments, Østby et al. [35] and later Jin et al. [36] study the interplay between astrocytic membrane mechanisms, transmembrane water movement, and extracellular space shrinkage in connection with neuronal activity. Neither of these latter models [35, 36] include a spatial dimension and thus do not account for intra-compartmental gradients. In pioneering work, Asgari and collaborators [37] study how astrocyte networks may modulate extracellular fluid flow and transport given a hydrostatic pressure difference between paraarterial and paravenous spaces via a spatially-discrete electrical analog model. A unifying framework fully incorporating spatial and temporal dynamics of

both ionic electrodiffusion and fluid movement in an arbitrary number of compartments is presented by Mori [38]. In this framework, hydrostatic and osmotic pressure gradients are assumed to drive fluid flow in both intra- and extracellular spaces. Zhu et al. [39] later extend this framework by including electro-osmosis as a driving force for interstitial fluid movement. They apply their framework to study the role of fluid flow on ionic transport in the optical nerve. However, little attention has been paid to quantifying the contributions of different driving forces for interstitial fluid flow during neuronal activity in the cortex.

Here, our target is two-fold: we aim to estimate the water movement induced by neuronal activity across astrocyte cell membranes, within astrocyte networks, and within the extracellular space, and to determine the relative role of different forces (osmotic, hydrostatic, and electrical) on ionic transport and fluid flow under such conditions. To estimate this electro-chemo-mechanical response, we introduce a high-fidelity computational model describing the spatial and temporal dynamics at the micro/milliscale of volume fractions, electrical potentials, ion concentrations, and hydrostatic pressures in an intracellular space (ICS) representing different astrocyte configurations and the extracellular space (ECS) (Fig 1). The model is embedded in the electrodiffusive Kirchhoff-Nernst-Planck framework and builds on previous work [40] incorporating ionic electrodiffusion [34, 38], fluid dynamics [39], and astrocyte modeling [34].

Our findings show that neuronal activity in the form of extracellular ionic input fluxes induces complex and strongly-coupled chemical-electrical-mechanical interactions in the astrocytic ICS and ECS. The response is characterized by membrane (electric) potential depolarization on the order of tens of millivolts and ECS water potentials ranging from a few to a hundred kilo-pascals, spatial differences in osmolarity on the order of several tens of millimolars, and fluid velocities ranging from a fraction—to tens of micrometers per minute. The fluid dynamics are crucially coupled to the spatial organization of the intracellular network. We observe intracellular fluid velocities in astrocyte networks of up to 14 μm/min, and fluid velocities in the ECS of similar magnitude. These velocities are dominated by an osmotic contribution in the intracellular compartment; without it, the estimated fluid velocities drop by a factor of ×34–45. Furthermore, the compartmental fluid flow has a pronounced effect on transport: advection accelerates ionic transport within astrocytic networks by a factor of ×1–5 compared to diffusion alone.

## 2 Results

In order to quantify the relative role of osmotic, hydrostatic, and electrical forces on transport and flow in cortical tissue, we ask the following questions. How do astrocyte and extracellular ion concentrations, electric potentials, pressures, and interstitial fluid velocities respond to changes in extracellular ion concentrations mirroring neural activity on the time scale of seconds? Moreover, to what extent do the mechanical responses (cellular swelling, fluid flow) contribute to alleviating ionic and mechanical ECS distress? To address these questions, we introduce a set of biophysical models for these quantities of interest, governed by the balance of mass, momentum, and charge, in combination with astrocyte membrane mechanisms in a representative volume.

### 2.1 A model for electrodiffusive, osmotic, and hydrostatic interplay in astrocyte networks

Ion- and fluid movement in an astrocyte network (ICS) and the extracellular space (ECS) is modeled via coupled partial differential equations (PDEs) in a homogenized model domain (Fig 1, Methods). Specifically, we consider a 1D domain of length 300 μm representing brain

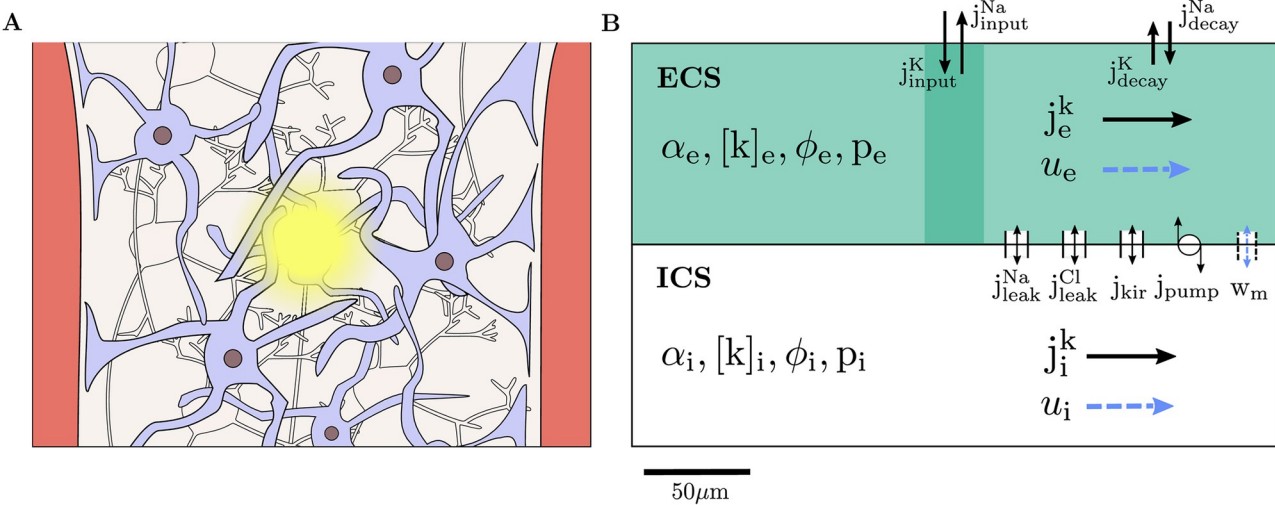

**Fig 1. Model schematics.** Illustration of brain tissue between two blood vessels with astrocytes (purple), neurons (grey), and ECS with neural activity in the center (**A**). The tissue is represented as a 1D domain of length 300 $\mu$m including ICS (astrocytes) and the ECS (**B**). Within each compartment, the model describes the dynamics of the volume fraction ($\alpha$), the Na$^+$, K$^+$, and Cl$^-$ concentrations ([Na$^+$], [K$^+$], [Cl$^-$]), the electrical potential ($\phi$), and the hydrostatic pressure ($p$). Neuronal activity is implicitly represented by K$^+$ and Na$^+$ input currents ($j_{input}^K$ and $j_{input}^{Na}$) in the input zone (of length 30 $\mu$m) and decay currents ($j_{decay}^K$ and $j_{decay}^{Na}$) across the whole domain. Transmembrane currents include an inward rectifying K$^+$ current ($j_{Kir}$), Na$^+$ and Cl$^-$ leak currents ($j_{leak}^K$ and $j_{leak}^{Cl}$), and a Na$^+$/K$^+$ pump current ($j_{pump}$). Intra- and extracellular currents ($j_i^k$ and $j_e^k$) are driven by electrodiffusion and advection. Fluid can travel across the membrane ($w_m$) and compartmentally in the intra- and extracellular space ($u_i$ and $u_e$).

tissue between two blood vessels, e.g., an arteriole and a venule. The model predicts the evolution in time and distribution in space of the volume fraction $\alpha_r$, the ion concentrations [Na$^+$]$_r$, [K$^+$]$_r$, and [Cl$^-$]$_r$, the electrical potential $\phi_r$, and the hydrostatic pressure $p_r$ in both ICS ($r$ = i) and ECS ($r$ = e). Ionic transport is driven by diffusion, electric drift, and advection. To model fluid movement in each compartment, we consider three different model scenarios:

**M1** The intra- and extracellular fluid flow is driven by hydrostatic pressure gradients. The astrocytic compartment can be interpreted as either a single closed cell or as a syncytium of cells without intercellular osmotic flow.

**M2** The intracellular fluid flow is driven by osmotic and hydrostatic pressure gradients, and the extracellular fluid flow is driven by the same mechanism as in M1. The astrocytic compartment can be interpreted as a syncytium of cells where osmosis acts as a driving force for fluid flow.

**M3** The intracellular fluid flow model is the same as in M2, and the extracellular fluid flow is driven by electro-osmosis in addition to hydrostatic pressure gradients. To include electro-osmosis as a driving force for ECS fluid flow is motivated by the narrowness of the ECS [39, 41].

For comparison, we also consider a zero-flow scenario [34]:

**M0** The compartmental fluid velocities and the transmembrane fluid flow (and thus cellular swelling) are assumed to be zero.

Transmembrane fluid flow in model scenarios M1–M3 is driven by hydrostatic and osmotic pressure differences. At the boundaries, we assume that no fluid and no ionic fluxes enter or leave the system. To account for transmembrane ionic movement, we include an

inward rectifying $K^+$ channel, passive $Na^+$ and $Cl^-$ channels, and a $Na^+/K^+$ pump. To ensure electroneutrality of the system, we include a set of immobile anions $a_r$. The immobile anions contribute to the osmotic pressures.

We mimic a scenario of high local neuronal activity by injecting a constant $K^+$ current into the ECS and simultaneously removing $Na^+$ ions in a stimulus zone in the middle of the computational domain. We set the strength of the input currents such that the extracellular $K^+$ concentration in the input zone reaches a maximum value of approximately 10 mM during the simulations. At this concentration level, we expect the $K^+$ buffering process to play a critical role; still, the concentration is below the level we observe in pathological conditions such as spreading depression (see Halnes et al. [34] and references therein). To maintain electroneutrality of the system, the $K^+$ and $Na^+$ input currents are of the same magnitude. Neuronal pumps and cotransporters are accounted for by removing $K^+$ ions from the ECS at a given decay rate and adding the same amount of $Na^+$ ions. The decay is proportional to the extracellular $K^+$ concentration and defined across the whole domain. We also study the effect of time-varying input by injecting pulsatile $K^+$ currents into the ECS. Note that the stimulus does not induce any osmotic pressure changes, as it does not affect the total osmotic concentration of the ECS.

## 2.2 Neuronal activity induces complex chemical-electrical-mechanical interplay

In order to understand and quantify the baseline electrical and mechanical response to chemical alterations, we first consider the model scenario where the compartmental fluid flow is only driven by hydrostatic pressure gradients (M1). Turning on the input currents at $t = 10$ s leads to changes in the ion concentrations, cellular swelling, depolarization of the glial membrane, and an increase in the transmembrane hydrostatic pressure difference. After about 40 seconds, the system reaches a new steady state, before all fields return to baseline levels after input offset at $t = 210$ s (Fig 2).

The input currents (Fig 2A) and the subsequent astrocytic activity lead to an increase of 6.68 mM in $[K^+]_e$, a decrease of 18.7 mM in $[Na^+]_e$, and a decrease of 16.9 mM in $[Cl^-]_e$, measured at the center of the input zone (Fig 2C). The increase in $[K^+]_e$ activates the $K^+$- and $Na^+$-decay currents (Fig 2B), which eventually lead the system back to baseline. Intracellularly, we observe an initial peak in $\Delta[K^+]_i$ of 3.19 mM before it settles on 0.428 mM. $[Na^+]_i$ and $[Cl^-]_i$ decrease by 6.44 mM and increase by 6.55 mM, respectively (Fig 2D). In response to the ionic shifts, the astrocytic compartment swells: the ICS volume increases by 13% (Fig 2F) and the ECS shrinks correspondingly by 26% (Fig 2E). As the initial size of the ECS is half that of the ICS, a change in ECS volume twice that of the ICS volume is as expected. Further, these volume changes affect the hydrostatic pressures: the transmembrane hydrostatic pressure difference increases by 118 Pa (Fig 2G). Finally, the glial membrane potential depolarizes from −86 mV to −61 mV (Fig 2H).

## 2.3 Transmembrane dynamics induce hydrostatic pressure gradients and compartmental fluid flow

Osmotically driven transport of fluid through AQP4, and possibly other membrane mechanisms, play an important role in cellular swelling and volume control of the ECS [42–45]. Whether cellular swelling induces hydrostatic pressure gradients driving compartmental fluid flow in the ICS and ECS is, however, far from settled [34]. We therefore also assess to what extent osmotic pressures induce hydrostatic pressures and fluid flow, still in the model scenario with hydrostatic-pressure-driven compartmental fluid flow (M1).

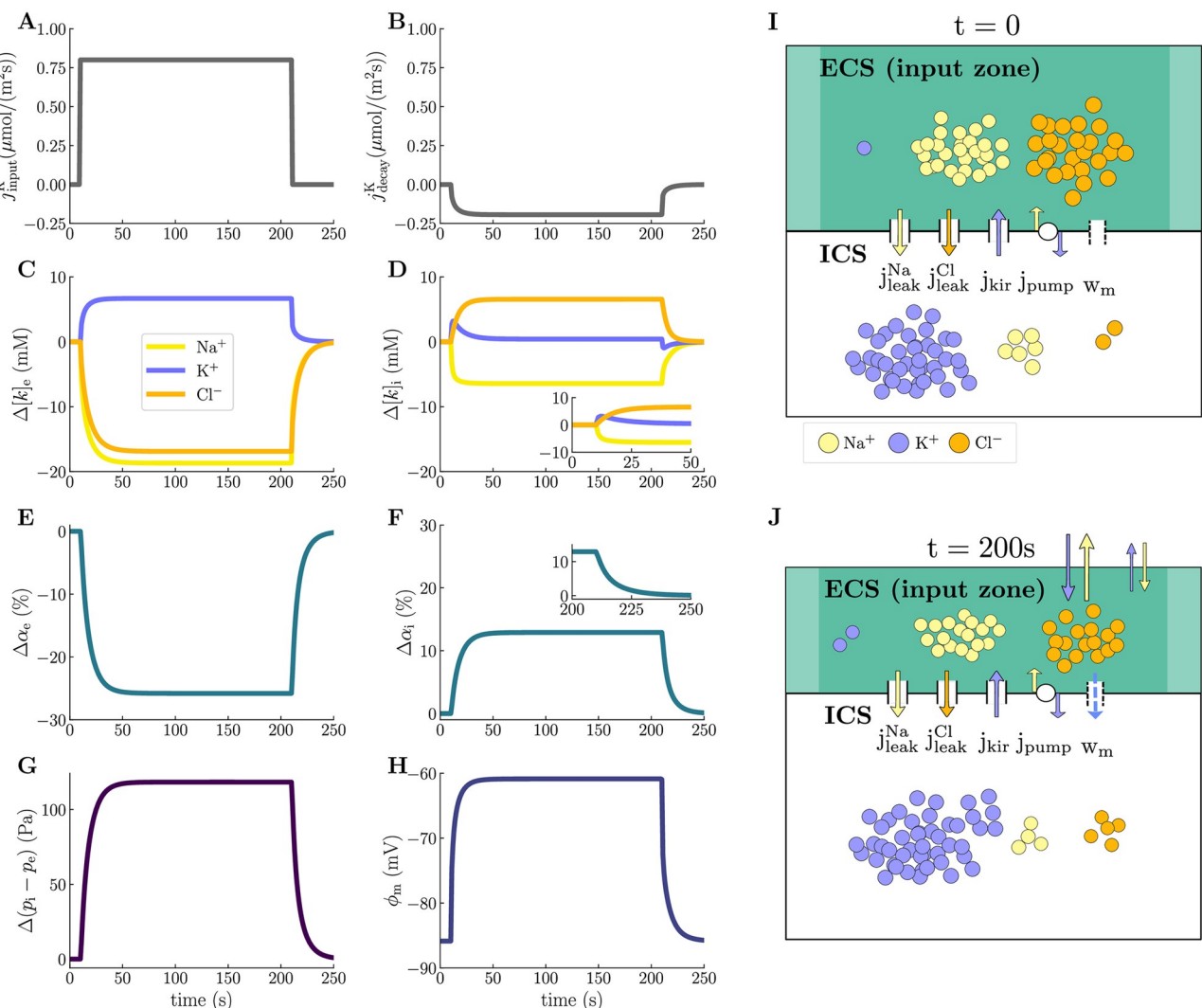

**Fig 2. Electrical, chemical, and mechanical dynamics in the input zone during local neuronal activity.** The panels display the time evolution of the K$^+$-injection current (**A**) and K$^+$-decay current (**B**), changes in the ECS (**C**) and ICS (**D**) ion concentrations, changes in the ECS (**E**) and ICS (**F**) volume fractions, changes in the transmembrane hydrostatic pressure difference (**G**), and membrane potential (**H**) at $x$ = 150 μm (center of the input zone). All changes are calculated from baseline values, which are listed in Methods. Panel **I** and **J** display a schematic overview of ionic dynamics and swelling at respectively $t$ = 0 s and $t$ = 200 s at $x$ = 150 μm.

The concentration shifts following astrocytic activity result in altered ICS and ECS osmolarities. Notably, the change in osmolarities peak in the input zone, where the intra- and extracellular osmolarities decrease by maximum 20.5 mM and 20.7 mM, respectively (Fig 3A). Consequently, the osmotic pressure across the astrocytic membrane decreases by a maximum of 713 Pa (Fig 3B). The osmotic pressure drives fluid across the astrocytic membrane, with a maximum velocity of 0.003 $\mu$m/min (Fig 3C), resulting in cellular swelling. Following swelling, the ICS hydrostatic pressure increases by at most 21.2 Pa in the input zone, while the ECS hydrostatic pressure drops by at most 97.1 Pa (Fig 3D). Note that the changes in compartmental hydrostatic pressures lead to a change in the transmembrane hydrostatic pressure gradient (Fig 3B), which affects the transmembrane fluid flow.

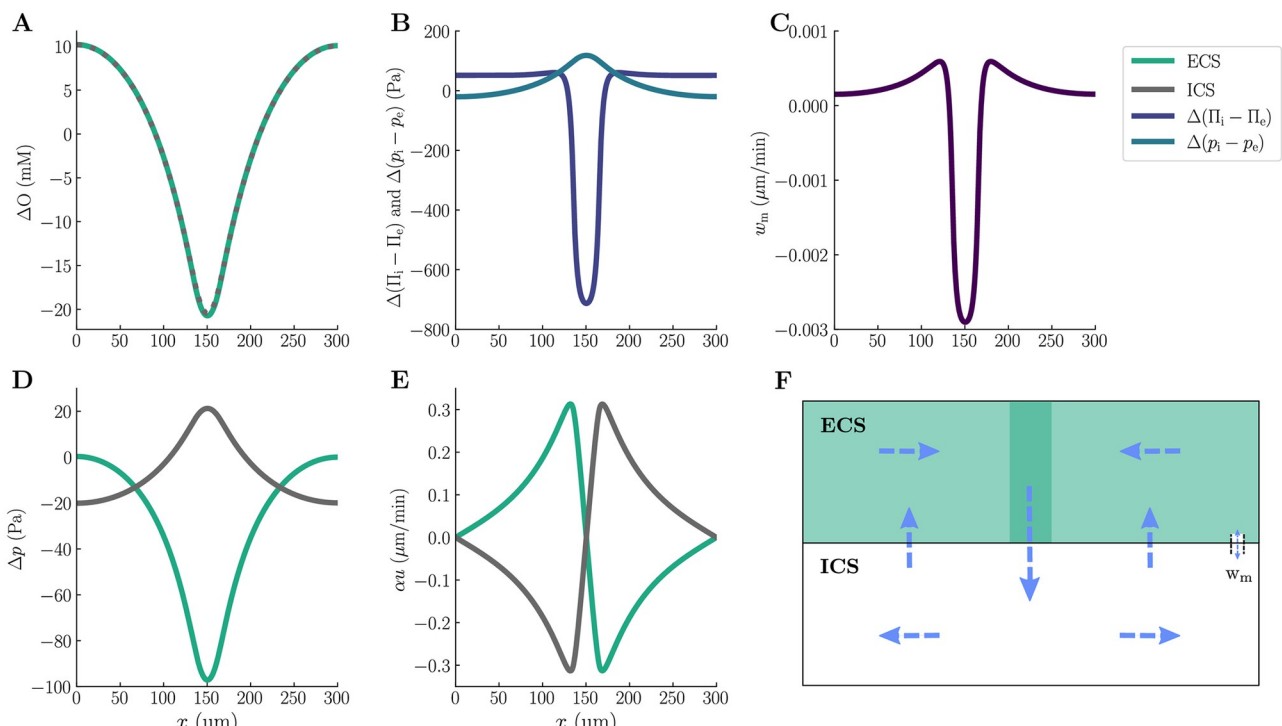

**Fig 3. Interplay between transmembrane- and compartmental pressures and fluid velocities (M1).** The panels display a snapshot (at $t = 200$ s) of the spatial distribution of the changes in intra- and extracellular osmolarities (**A**), osmotic and hydrostatic pressure gradients across the glial membrane (**B**), transmembrane fluid velocity (**C**), changes in the intra- and extracellular hydrostatic pressures (**D**), intra- and extracellular superficial fluid velocities (**E**), and illustration of the flow pattern (**F**). All changes are calculated from baseline values, which are listed in Methods.

The ICS and ECS hydrostatic pressure gradients drive compartmental fluid flow forming two circulation zones (Fig 3E and 3F). The intra- and extracellular superficial fluid velocities, $\alpha_r u_r$, peak at 0.31 $\mu$m/min (Fig 3E). Note that the two superficial fluid velocities are opposite in direction but have the same magnitude—this is a direct consequence of the incompressibility condition and no-flux boundary conditions in one dimension. During steady-state and cellular swelling in the input zone, fluid flows across the membrane into the ICS (Fig 3C). In the ICS, the fluid consequently flows away from the input zone, with positive flow to the right of the swelling and negative flow to the left of the swelling (Fig 3E). The ECS fluid flow is in the opposite direction, towards the input zone (Fig 3E).

## 2.4 Extracellular and intracellular fluid flow alleviate osmotic pressure build-up

In a previous study [46], we predicted the strength of osmotic pressure build-up across an astrocyte membrane using a classical electrodiffusive model not accounting for fluid flow. However, to what extent will swelling and compartmental fluid flow affect osmotic pressure build-up across the membrane? Here, we investigate this question by comparing predictions of the zero-flow model (M0), the flow model without intercellular osmotic flow (M1), and the flow model with osmotic intercellular flow (M2).

The ICS and ECS osmolarities are altered by astrocytic activity in all model scenarios, notably peaking in the input zone (Fig 4A and 4B). In the ECS, the osmolarity decrease by 36.5 mM, 20.7 mM, and 6.74 mM for M0, M1, and M2, respectively (Fig 4A). Interestingly,

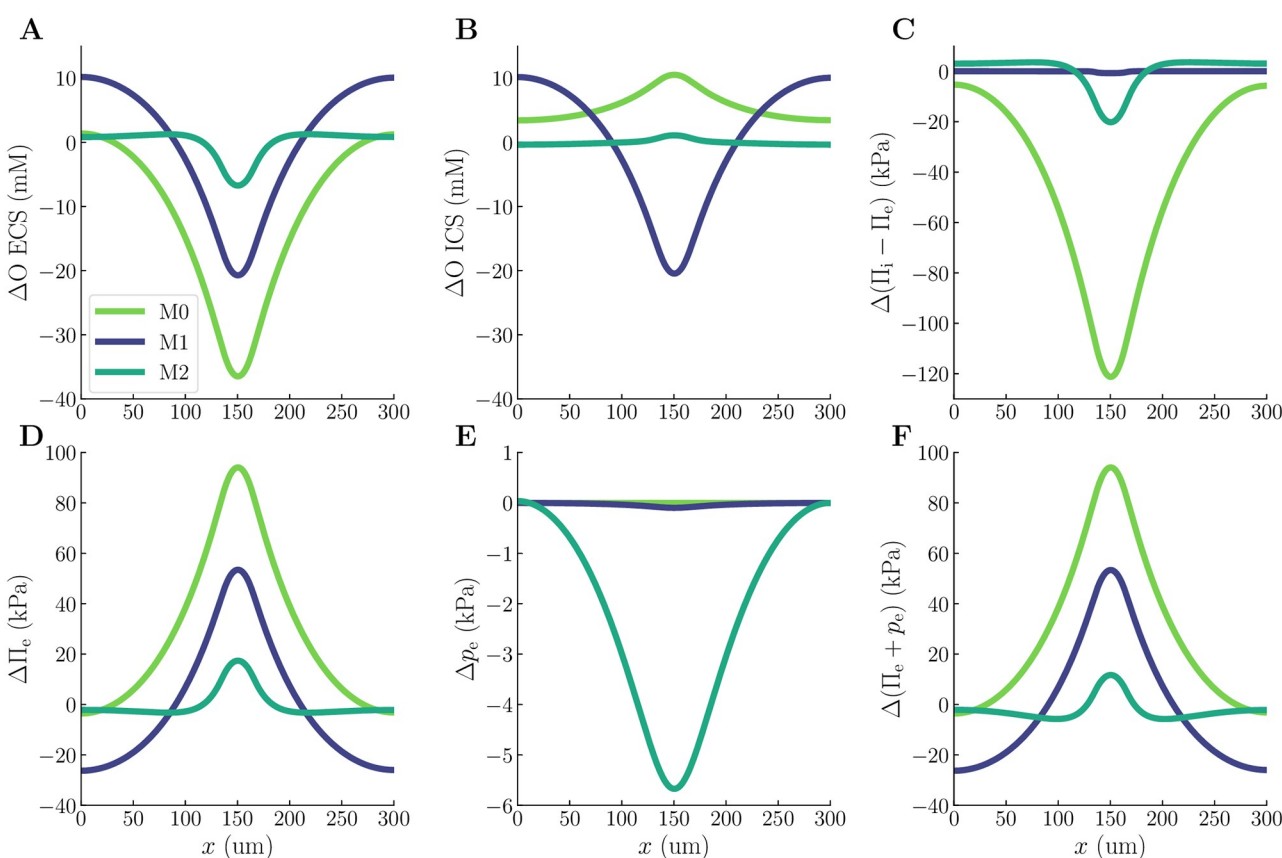

**Fig 4. Comparison of osmotic pressures and ECS water potentials predicted by model scenarios M0, M1, and M2.** The upper panels display a snapshot of ICS osmolarities (**A**), ECS osmolarities (**B**), and osmotic pressures across the membrane (**C**). The lower panels display a snapshot of ECS solute potentials (**D**), ECS hydrostatic pressures (**E**), and ECS water potentials (**F**). All panels display the deviation from baseline levels at $t = 200$ s.

the ICS osmolarity increases by 10.5 mM and 1.09 mM for M0 and M2, respectively, whereas it decreases for model scenario M1 by 20.45 mM (Fig 4B). The decrease in ICS osmolarity in model scenario M1 results from cellular swelling: the osmolarity is defined as the amount of ions per unit volume. An increase in cell volume may thus cause the ion concentration to drop even if the number of ions increases. Furthermore, we can convert the intra- and extracellular osmolarities to intra- and extracellular solute potentials, $\Pi_i$ and $\Pi_e$, respectively (see Section 4.2 for further details). Taking the difference in solute potential across the membrane gives us the osmotic pressure, which differs substantially between the models: M0, M1, and M2 predict a maximum drop in osmotic pressure of respectively 121 kPa, 0.713 kPa, and 20.2 kPa (Fig 4C). Allowing for cellular swelling and compartmental fluid flow thus reduces the osmotic pressure across the membrane by 99.4% (M1) and 83.3% (M2). These findings suggest that model scenario M0, or generally any model for electrodiffusion not taking into account fluid dynamics, highly overestimates the osmotic pressure building up across the membrane.

## 2.5 Volume dynamics is essential for ECS homeostasis

Water uptake in astrocytes via e.g. AQP4 has been hypothesized to contribute to stabilizing ECS ion concentrations, and thus prevent severe neuronal swelling. Swelling is driven by

the difference in intra- and extracellular water potential, which is given by the solute potential $\Pi_r$ plus the hydrostatic pressure $p_r$ (see Methods for details). An increase in extracellular water potential will result in neuronal swelling as water flows along the potential gradient.

For all model scenarios (M0, M1, M2), the ECS solute potential increases (Fig 4D). The increase is most severe for the zero-flow model (M0), which predicts a maximum increase of 94.1 kPa. When taking swelling and compartmental hydrostatic-pressure-driven fluid flow into account (M1), the ECS solute potential increases by maximum 53.5 kPa, whereas adding ICS osmotic forces (M2) leads to an increase of maximum 17.4 kPa. In M0, the change in $p_e$ is zero by definition, whereas M1 predicts a negligible maximal drop (0.0971 kPa, Fig 4E). Conversely, model scenario M2 predicts a pronounced hydrostatic pressure drop (5.67 kPa, Fig 4E). Consequently, the maximal change in the ECS water potential is substantially smaller in M2 (11.7 kPa, Fig 4F) than in M0 and M1 (respectively 94.1 kPa and 53.4 kPa, Fig 4F). The maximal change in the ECS solute potential is less severe for M2 than for M0 and M1 (Fig 4D). Additionally, while the ECS solute potentials increase (Fig 4D), the ECS hydrostatic pressures decrease (Fig 4E), and thus drive the water potential in opposite directions. Consequently, the ECS hydrostatic pressure change in M2 reduces the contribution from the less severe change in ECS solute potential, resulting in a lower ECS water potential. Thus, cellular swelling and osmotic transport within the astrocytic network (M2) contribute to prevent water potential build-up in the ECS.

## 2.6 Astrocyte osmotics strengthens compartmental fluid flow

The presence of driving forces for fluid flow—both at the brain microscale and organ-level—remain an open question. To assess the potential contributions from osmosis in the astrocytic network (M2) and electro-osmosis in the ECS (M3), we here compare the compartmental fluid velocities, cellular swelling, and hydrostatic pressures predicted by model scenarios M1, M2, and M3.

The maximum superficial fluid velocity predicted by M2 is 14 $\mu$m/min—about 45 times larger than for M1 (Fig 5A and 5B and Table 1). For M3, the maximum superficial fluid velocity is 13 $\mu$m/min—slightly smaller than for M2 (Fig 5C and 5D and Table 1). For both M2 and M3, the fluid velocities are dominated by osmosis in the ICS (Fig 5A and 5C) and hydrostatic forces in the ECS (Fig 5B and 5D). Interestingly, the intracellular hydrostatic pressure gradient drives fluid towards the input zone in M2 and M3 (Fig 5A and 5C), as opposed to the intracellular hydrostatic pressure in M1 which drives fluid away from the input zone (Fig 3C). The difference in ICS flow direction predicted by M1, M2, and M3 arises from the coupling of the hydrostatic-, osmotic-, and electro-osmotic forces in the mathematical model: the osmotic- and electro-osmotic forces are given by the ion concentration- and electrical potential gradients, respectively, whereas the hydrostatic pressure result from the incompressibility of the interstitial fluid (cf. Eq (7)). Less cellular swelling is predicted by M2 and M3 than by M1: the astrocyte swells by respectively 12.9%, 3.74%, and 4.55% in M1, M2, and M3 (Table 1). Finally, we observe notable differences in the intra- and extracellular hydrostatic pressures: the maximum ICS hydrostatic pressure is 1.02 kPa, −4.64 kPa, and −10.3 kPa in M1, M2, and M3, respectively (Table 1). The maximum ECS hydrostatic pressure is -0.0971 kPa, -5.67 kPa, and -11.3 kPa in M1, M2, and M3, respectively (Table 1). The transmembrane hydrostatic pressures are similar in M2 and M3 even if the intra- and extracellular pressures are different: the maximum transmembrane hydrostatic pressure is respectively 1.03 kPa and 1.04 kPa in M2 and M3 (Table 1). The maximum transmembrane hydrostatic pressure in M1 is 1.12 kPa (Table 1).

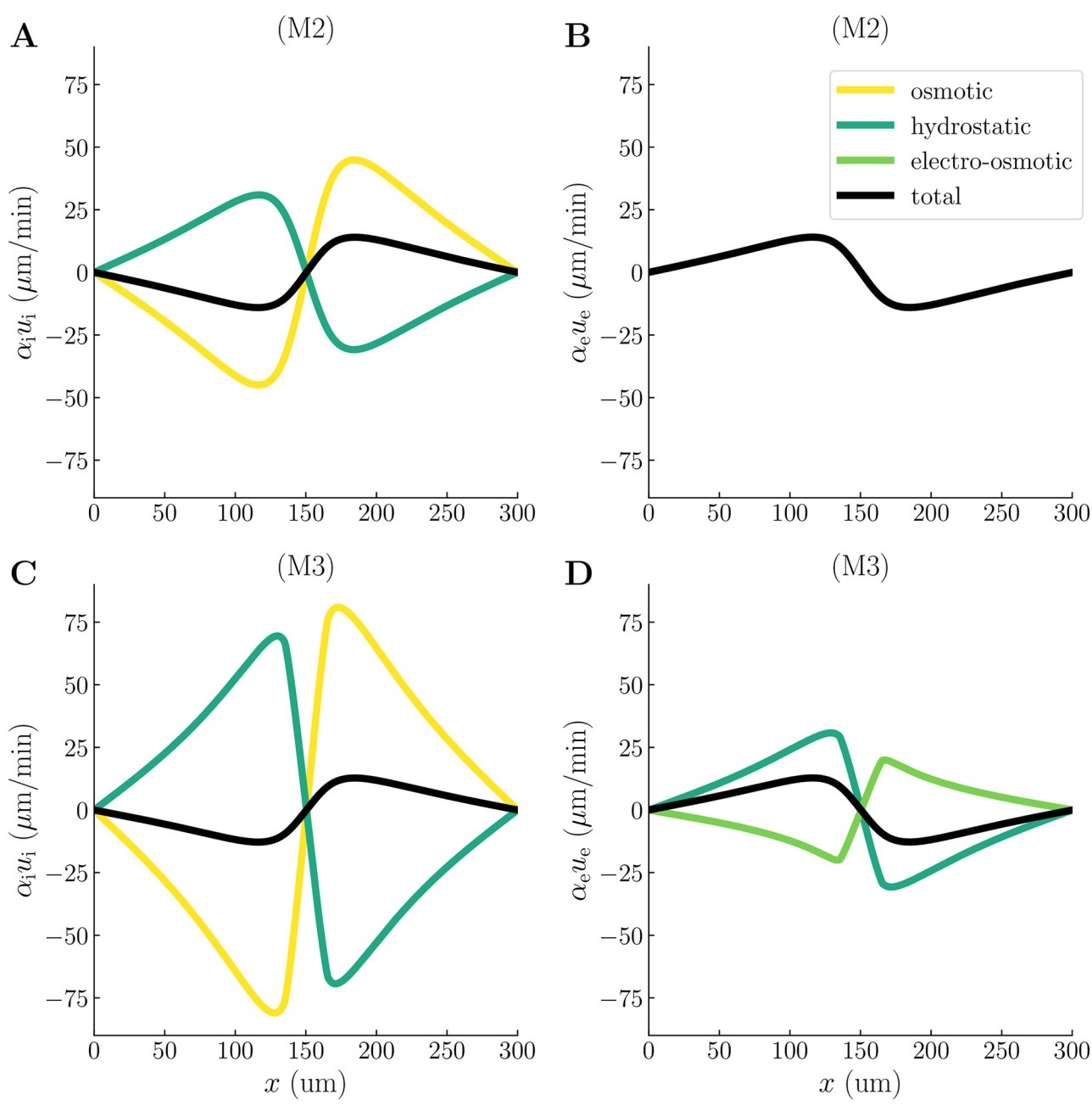

**Fig 5. Fluid velocities predicted by model scenarios M2 and M3.** Spatial profiles of the total superficial fluid velocities (black dashed lines), together with their hydrostatic (dotted green lines), osmotic (yellow line), and electro-osmotic (solid green line) contributions at $t = 200$ s. The upper panels show the intra- **(A)** and extracellular **(B)** fluid velocities for model scenario M2. The lower panels show the intra- **(C)** and extracellular **(D)** fluid velocities for model scenario M3.

## 2.7 Astrocyte osmotics accelerates ionic transport and alters role of advection

In model scenarios M1–M3, the compartmental fluid velocities will contribute to ionic transport via advection. To assess the role of advection in compartmental ionic transport, we compare model scenarios M1 and M3. Specifically, we decompose the intra- and extracellular ionic

**Table 1. Quantities of interest for the different model scenarios (M0, M1, M2, and M3).**

| Model scenario | M0 | M1 | M2 | M3 |
|---|---|---|---|---|
| ICS swelling (%) | – | 12.9 | 3.74 | 4.55 |
| ECS shrinkage (%) | – | 25.8 | 7.48 | 9.12 |
| ICS osmolarity (mM) | 315 | 283.9 | 305 | 303 |
| ECS osmolarity (mM) | 268 | 283.3 | 297 | 296 |
| Osmotic pressure (kPa) | −122 | −1.71 | −21.2 | −19.4 |
| ICS hydrostatic pressure (kPa) | – | 1.02 | −4.64 | −10.3 |
| ECS hydrostatic pressure (kPa) | – | −0.0971 | −5.67 | −11.3 |
| Transmembrane hydrostatic pressure difference (kPa) | – | 1.12 | 1.03 | 1.04 |
| max ICS intrinsic fluid velocity (μm/min) | – | 0.71 | 34 | 31 |
| max ECS intrinsic fluid velocity (μm/min) | – | 2.0 | 75 | 69 |
| max ICS superficial fluid velocity (μm/min) | – | 0.31 | 14 | 13 |
| max ECS superficial fluid velocity (μm/min) | – | 0.31 | 14 | 13 |
| Transmembrane fluid velocity (μm/min) | – | −0.0029 | −0.099 | −0.090 |

All measurements are taken at $t = 200$ s and at $x = 150$ μm, except from the max ICS- and ECS intrinsic fluid velocities ($u_r$) and the max ICS- and ECS superficial fluid velocities ($\alpha_r u_r$) which are the maximum values over space at $t = 200$ s. (–) denotes that the value is not applicable.

fluxes and calculate the advection/diffusion fraction ($F_{diff}$) and the advection/drift fraction ($F_{drift}$) for each of the ionic species (see Methods for further details).

We observe that $F_{diff}$ and $F_{drift}$ range from 0.002 to 0.062 in model scenario M1, indicating that advection plays a negligible role in ionic transport (Fig 6A–6F). In model scenario M3, however, we observe a larger variability in the advection/diffusion- and advection/drift rates: $F_{diff}$ ranges from 0.058 to 4.519, and $F_{drift}$ ranges from 0.325 to 0.949 (Fig 6G–6L). For K$^+$ transport in the M3 model, the advective flux is on the same order of magnitude as the intra- and extracellular electric drift and about 4.5 times stronger than the intracellular diffusion ($F_{diff} = 4.519$, Fig 6G and 6J). Advection plays the most important role intracellularly and accelerates the K$^+$ transport; For M1, the intracellular K$^+$ flux has a maximum value of 58 $\mu$mol/(m$^2$s) (Fig 6D), whereas for M3, the maximum intracellular K$^+$ flux is 70 $\mu$mol/(m$^2$s) (Fig 6J). For Na$^+$ and Cl$^-$ transport in the M3 model, advection is on the same order of magnitude as diffusion and electric drift (Fig 6H–6I and 6K–6L). The advection even dominates intracellular diffusion of Na$^+$ ($F_{diff} = 1.286$, Fig 6K) and extracellular diffusion of Cl$^-$ ($F_{diff} = 4.079$, Fig 6I). Overall, advection accelerates the transport of total charge in the system: For M1, the charge flux (defined here as $z_K j_K + z_{Na} j_{Na} + z_{Cl} j_{Cl}$) is maximum 59 $\mu$mol/(m$^2$s), whereas for M3, the charge flux is maximum 71 $\mu$mol/(m$^2$s).

## 2.8 Fluid flow patterns induced by low-frequency activity waves

Till now, we have studied the quasi-static response resulting from a step-wise change in (constant) input fluxes. To study how low-frequency waves of ionic changes affect the osmotic, hydrostatic, and electrical response of the system, we compare three different stimulus protocols via model scenario M3: (I) (slow) input fluxes varying at a low (1Hz) frequency, (II) (ultraslow) input fluxes varying at an ultralow (0.05 Hz) frequency, and (III) constant input fluxes, with on- and offset times $t = 10$ s and $t = 210$ s for all protocols and same amplitude for all input currents (Fig 7A–7C, see Methods for further details).

The ECS potential, ECS K$^+$ concentration, and ECS volume fraction reach periodic steady states for both the slow and ultraslow stimuli (Fig 7D, 7G, 7J, 7E, 7H and 7K,

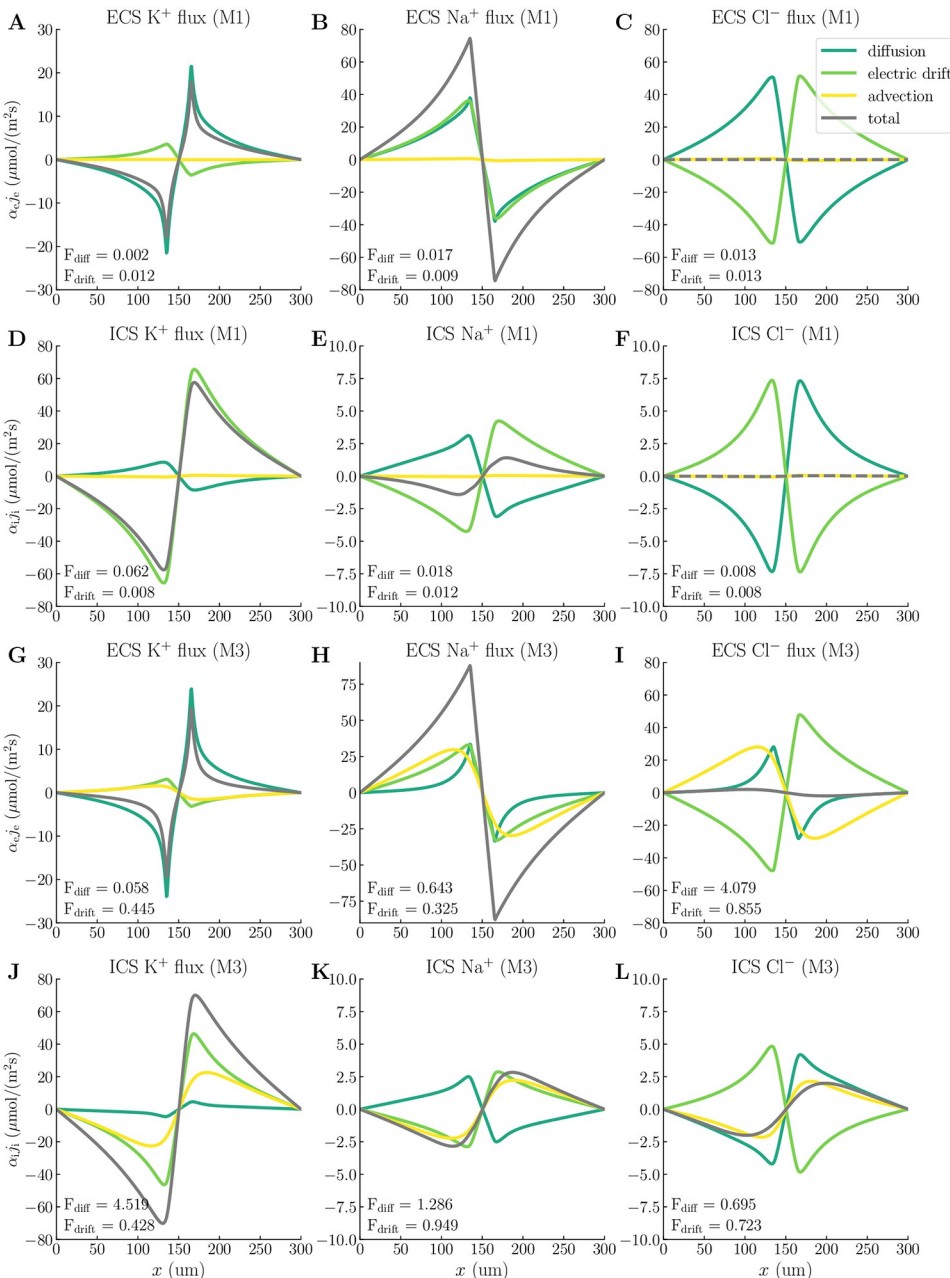

**Fig 6. Compartmental ionic fluxes.** Spatial profiles of the total compartmental ionic fluxes (grey dashed lines), and their diffusive (dark green lines), electric drift (light green lines), and advective (yellow lines) components at $t = 200s$ for the different ionic species. Each panel additionally contains the advection/diffusion fraction ($F_{\text{diff}}$) and the advection/electric drift fraction ($F_{\text{drift}}$) for the associated ion species. Panels **A-F** display fluxes for modeling scenario M1, and panels **G-L** display fluxes for modeling scenario M3. All fluxes are multiplied by the volume fraction $\alpha$.

respectively). Interestingly, we observe phase shifts in $\phi_e$, $\alpha_e$, and $[\text{K}^+]_e$ compared to the input flux. The phase shifts are more pronounced for the slow than the ultraslow stimuli. As expected, when $j^K_{\text{input}}$ increases, $\phi_e$ and $\alpha_e$ decrease, while $[\text{K}^+]_e$ increases on average. The phase of $[\text{K}^+]_e$ is shifted by 10% and 5.0% (relative to the cycle period and the input wave) for the slow and ultraslow stimuli, respectively, with the corresponding numbers

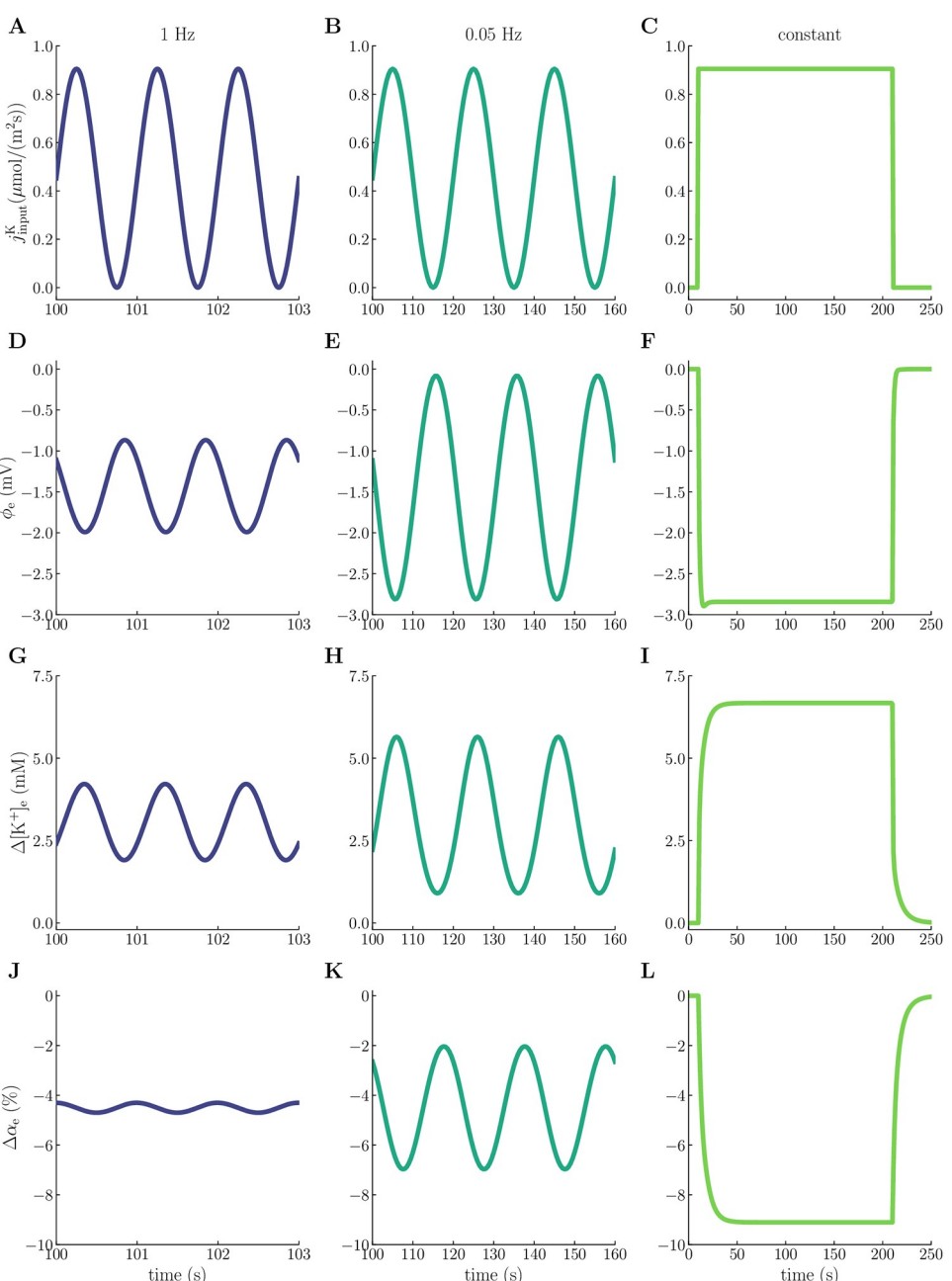

**Fig 7. Electrical, chemical, and mechanical dynamics during slow, ultraslow, and constant stimuli.** The panels display the time evolution of the $K^+$-input current (**A,B,C**), extracellular potential (**D,E,F**), changes in the ECS $K^+$ concentration (**G,H,I**), and changes in the ECS volume fraction (**J,K,L**), at $x = 150$ μm (center of the input zone) for input varying at 1 Hz (left column), input varying at 0.05 Hz (middle column), and constant input (right column) (see Methods for details). All simulations correspond to model scenario M3.

being 60% and 53% for $\phi_e$ and 74% and 63% for $\alpha_e$ (Fig 7A–7B, 7D–7E, 7G–7H and 7J–7K). The amplitudes of $\phi_e$, $[K^+]_e$, and $\alpha_e$ are smaller for the slow stimulus (Fig 7D, 7G and 7J) than the ultraslow (Fig 7E, 7H and 7K) and constant (Fig 7F, 7I and 7L) stimuli. For the ultraslow stimulus, the amplitude of $\phi_e$ is the same as for the constant stimulus (Fig 7E and 7F), whereas the amplitudes of $[K^+]_e$ and $\alpha_e$ are smaller compared to the constant

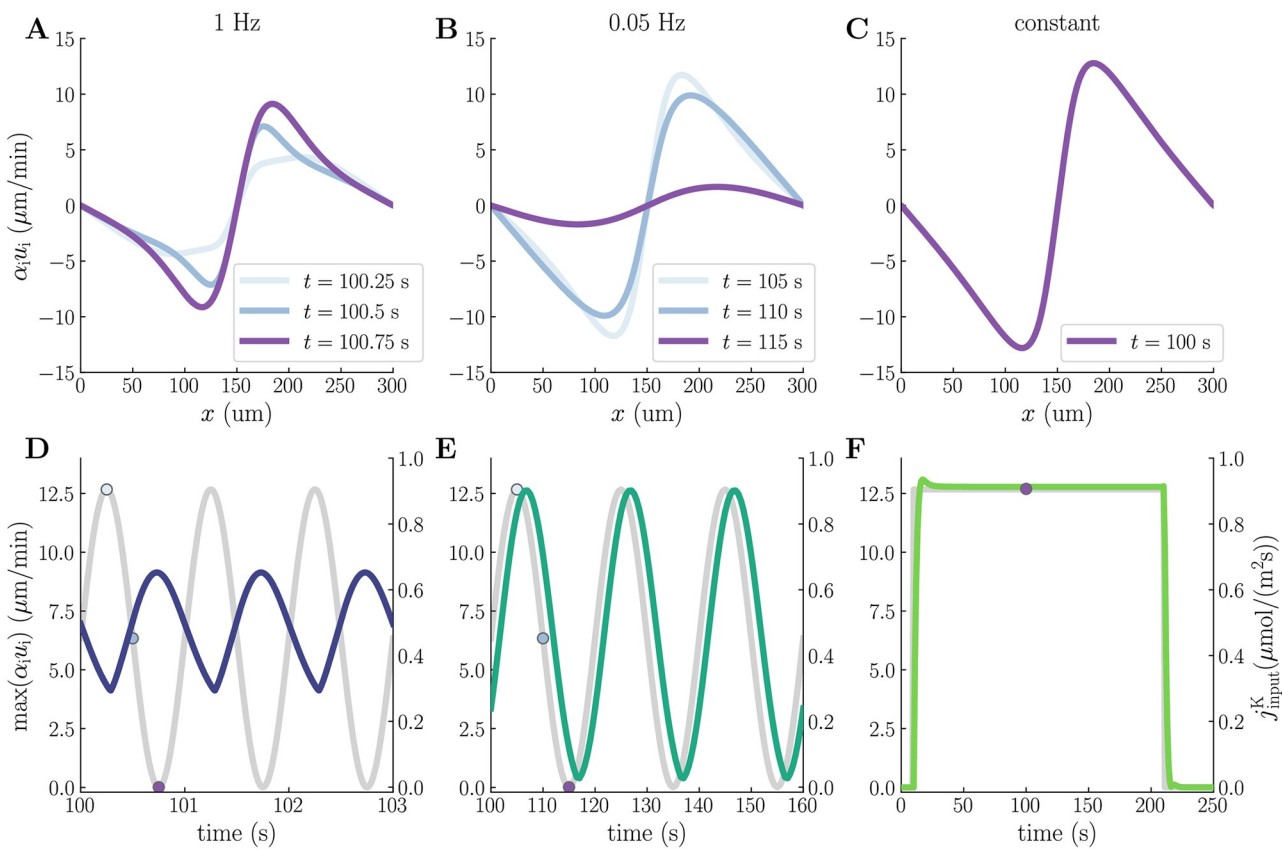

**Fig 8. Compartmental fluid velocities during slow, ultraslow, and constant stimuli.** The upper panels display spatial profiles of the intracellular superficial fluid velocities at peak (light blue), average (blue), and nadir input (purple) for slow (1 Hz) (**A**), ultraslow (0.05 Hz) (**B**), and constant (**C**) stimuli. The lower panels display the time evolution of the maximum intracellular superficial fluid velocity for the slow (**D**), ultraslow (**E**), and constant (**F**) stimuli, plotted alongside the corresponding K⁺-input current (gray). The dotted markers in panels **D**–**F** correspond to the time points in panels **A**–**C**. Note that panels **D**–**F** show different time windows on the x-axes. All simulations were run for 250 s and correspond to model scenario M3.

stimulus (Fig 7H–7I and 7K–7L). These reduced amplitudes are a consequence of the lower total input flux for the pulsatile stimuli, with a more substantial reduction for the slow stimulus.

The slow and ultraslow stimuli also cause periodic variations in the superficial fluid velocities, as opposed to the constant input, which causes the maximum superficial fluid velocity to plateau at 13 μm/min (Fig 8). The slow stimulus induces maximum superficial fluid velocities varying between 4.1 μm/min and 9.1 μm/min (Fig 8D), whereas the ultraslow stimulus causes the maximum superficial fluid velocity to vary between 0.39 μm/min and 13 μm/min (Fig 8E). The maximum superficial fluid velocity peaks near nadir input for the slow stimulus (Fig 8D), and near peak input for the ultraslow stimulus (Fig 8E).

## 2.9 Flow sensitivity to changes in permeabilities, stiffness, and input flux density

The magnitude of the compartmental fluid velocities $\alpha \cdot u$ is likely to depend on the choice of parameters. We thus perform a sensitivity analysis where we measure the maximum superficial fluid velocity for different values of the compartmental permeability $\kappa$, the membrane stiffness $K_m$, and the membrane water permeability $\eta_m$ (Fig 9). Specifically, we compare the sensitivity

of modeling setups M1 and M3 in order to assess whether the differences we have observed between the two are robust with respect to the parameter choices.

By increasing $\kappa$ from 0.0 m²/(Pa s) to $1 \cdot 10^{-12}$ m²/(Pa s) (with the default value set to $1.8375 \cdot 10^{-14}$ m²/(Pa s)), the superficial fluid velocity increases from 0.0 μm/min to 7.1 μm/

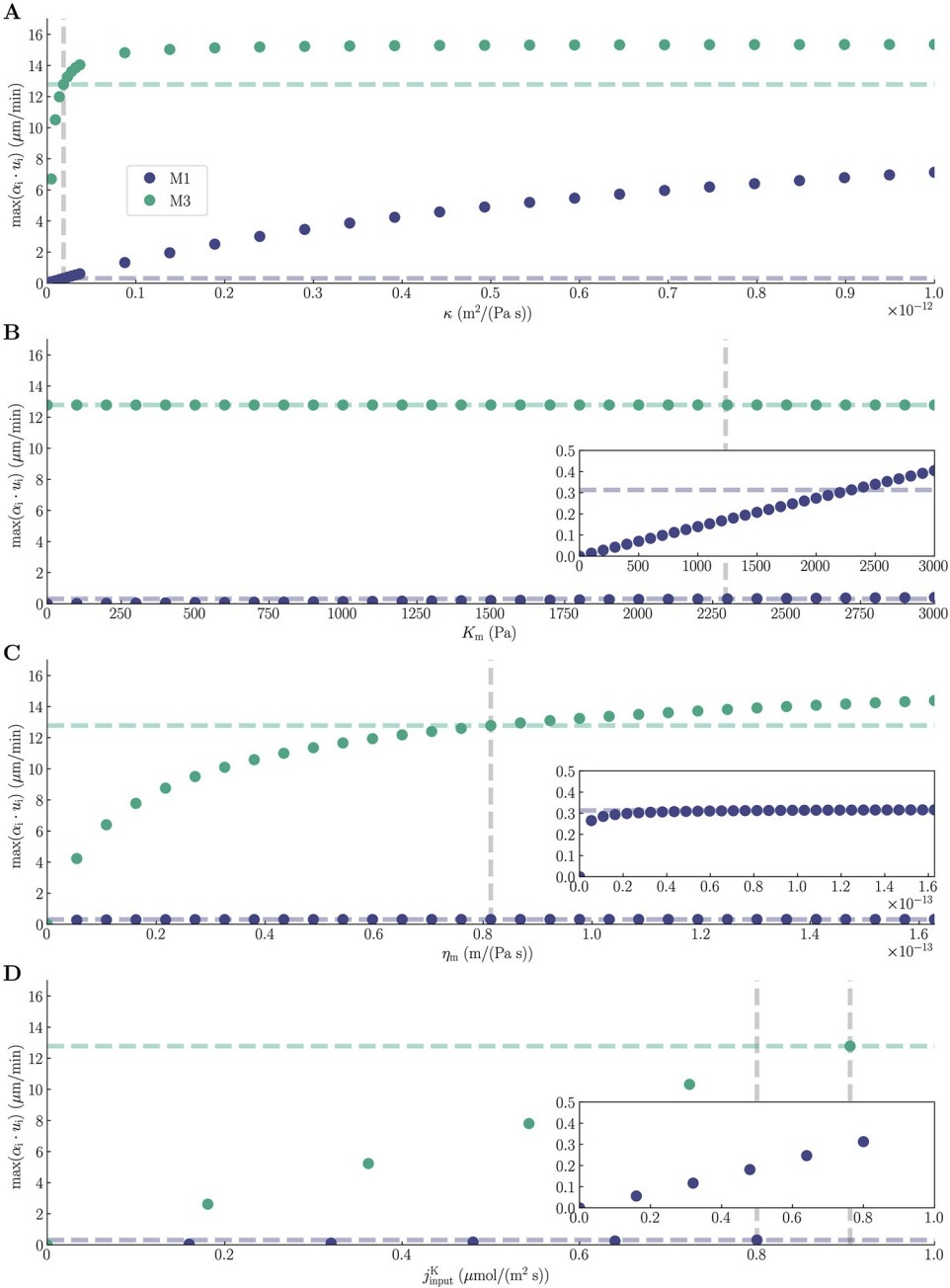

**Fig 9. Sensitivity analysis.** The maximum superficial fluid velocity, $\alpha_i u_i$, for different values of the compartmental permeability $\kappa$ (**A**), membrane stiffness $K_m$ (**B**), membrane water permeability $\eta_m$ (**C**), and input flux density $j^K_{input}$ (**D**) at $t = 200$ s for modeling scenarios M1 (blue dots) and M3 (green dots). The horizontal dashed lines mark the maximum value of $\alpha_i u_i$ corresponding to the default values of the model parameters, marked with vertical dashed lines. The default value of $\kappa$ was set to be the same in the ICS and ECS, and we changed the two simultaneously by the same amount.

min for M1 and from 0.0 μm/min to 15 μm/min for M3 (Fig 9A). The superficial fluid velocity in M3 converges around $\kappa = 1.4 \cdot 10^{-13}$ m²/(Pas), while the superficial fluid velocity in M1 continues to increase through $\kappa = 1 \cdot 10^{-12}$ m²/(Pas). The difference between M1 and M3 in superficial fluid velocities decreases for $\kappa$ values above the default parameter choice, but we still observe a notable difference of 215% at $\kappa = 1 \cdot 10^{-12}$ m²/(Pas). For $\kappa$ values below the default parameter choice, the absolute difference between M1 and M3 fluid velocities decreases, but the relative difference increases. Increasing $K_m$ from 0.0 Pa to 3000 Pa (with the default value set to 2294 Pa) increases the superficial fluid velocity linearly from 0.0 μm/min to 0.40 μm/min for M1 (Fig 9B). For M3, the superficial fluid velocity is 13 μm/min for all values of $K_m$. The small changes in M1 fluid velocities and constant M3 fluid velocity lead to a close-to-constant absolute difference between the predictions made by the two modeling setups when we vary $K_m$. By increasing the membrane water permeability $\eta_m$ from 0.0m/(Pas) to $1.628 \cdot 10^{-13}$ m/(Pas) (with the default value set to $8.14 \cdot 10^{-14}$ m/(Pas)), the superficial fluid velocity increases from 0.0 μm/min to 0.32 μm/min for M1 and from 0.0 μm/min to 14 μm/min for M3 (Fig 9C). The difference in superficial fluid velocity between M1 and M3 decreases for $\eta_m$ values below the default choice, but we still observe that M3 predicts a 16 times higher superficial fluid velocity than M1 (4.2 μm/min vs. 0.27 μm/min) for $\eta_m$ as low as $5.43 \cdot 10^{-15}$ m/(Pas). To study how changes in the input flux density $j_{input}^K$ affect the superficial fluid velocities, we run the simulations using 20%, 40%, 60%, 80%, and 100% of the default input flux value ($8.0 \cdot 10^{-7}$ mol/(m²s) for M1 and $9.05 \cdot 10^{-7}$ mol/(m²s) for M3). Both modeling setups show a linear relationship between $j_{input}^K$ and the superficial fluid velocities (Fig 9D). The absolute difference between M1 and M3 decreases when $j_{input}^K$ decreases, but M3 still predicts a superficial fluid velocity that is 47 times higher than the prediction made by M1 when the input flux density is set to 20% of the default value.

## 3 Discussion

Our results demonstrate that localized extracellular K⁺ influx in conjunction with Na⁺ efflux, reflecting a zone of high neuronal activity, induces a strongly coupled and complex chemical-electrical-mechanical response in astrocyte ICS and the ECS with spatial and temporal changes in osmolarities, swelling, electrical potentials, pressures, and fluid flow. Cellular swelling and, importantly, osmotically driven fluid flow within the astrocytic network contribute to preventing high levels of extracellular water potential, effectively protecting against neuronal swelling. Fluid flow within each compartment may reach tens of μm/min and, as such, substantially contributes to the overall dynamics. Compartmental fluid flow, in concert with cellular swelling, alleviates osmotic pressure build-up and accelerates ionic transport within astrocytic networks by a factor of ×1–5 compared to diffusion alone.

The shifts in ECS ion concentrations are in line with experimental observations and reports from comparable modeling studies. Experimentally, ECS K⁺ is measured to increase by 6—12 mM during sensory stimulation and strong electrical stimulation [2]. Halnes et al. [34, 47], Østby et al. [35], and Sætra et al. [48] report an increase in ECS K⁺ in the range of 5—10 mM via *in-silico* studies. We observe an increase in ECS K⁺ of 6.68 mM in the input zone during stimuli. Further, we observe a decrease in both ECS Na⁺ and Cl⁻ concentrations, which is in agreement with previous modeling studies [34, 47]. Interestingly, we observe a decrease in the ICS Na⁺ concentration: although the number of Na⁺ ions increases, the increase in ICS volume results in a total decrease in concentration. Our findings suggest that both ICS Cl⁻ and K⁺ concentrations increase during neuronal activity. This is in agreement with previous experimental reports [49] and with K⁺ uptake in astrocytes facilitating clearance of excess interstitial K⁺ following neuronal activity [2].

Astrocytes swell in response to $K^+$ influx, reducing the interstitial space volume by up to 30% [42, 50, 51]. The local interplay between astrocytic $K^+$ uptake and ECS shrinkage has previously been studied computationally using single neuron-glia-ECS unit models, not considering spatial buffering [35, 36]. Østby et al. [35] report that inclusion of the co-transporters NBC and NKCC1, together with NKCC1-dependent water transport, is necessary to obtain ECS shrinkage that matches experimentally observed values. In their model scenario including only the major basic membrane processes (i.e., $Na^+/K^+$ pump, passive ion transport, and osmotically driven water transport), they obtain an ECS shrinkage of 10.8 ± 4.0%. In contrast, Jin et al. [36] report that their model accounts for experimental observations without including non-AQP4 water transport pathways. We observe a 25% shrinkage of the ECS with only passive transmembrane water transport during stimuli comparable to that of Østby et al. [35] and Jin et al. [36]

While the importance of osmotic effects has been widely recognized in the context of interstitial fluid flow and production, it has remained and remains hard to quantify [3, 16]. When prescribing a hydrostatic pressure difference of 1 mmHg/mm but ignoring electrochemical contributions and interactions between the ECS and ICS, Holter et al. [27] arrived at a superficial ECS fluid velocity estimate of less than 0.2 μm/min. Here, accounting for the combined biophysical effects of ionic electrodiffusion, cellular swelling, and fluid flow by hydrostatic and osmotic pressures, we estimate that neuronal activity may induce transmembrane fluid velocities on the order of 0.1 μm/min, intracellular fluid velocities in astrocyte networks of up to 14 μm/min, and fluid velocities in the ECS of similar magnitude. These velocities are dominated by an osmotic contribution in the intracellular compartment; without it, the estimated fluid velocities drop by a factor of ×34–45. Our estimates are very much in line with the interstitial bulk flow velocities of 5.5–14.5 μm/min [52–54], 10.5 μm/min [55], or 10.0 μm/min [56], as reported by Nicholson [57], the average interstitial bulk velocities in humans of 1–10 μm/min as quantified by Vinje et al. [32], and in the lower range of the 7–50 μm/min bulk flow velocities identified by Ray et al [58]. Comparing with the pioneering modeling study by Asgari et al. [37], they report baseline flow estimates resulting from a hydrostatic pressure difference (of unknown origin) alone of $\approx 1\text{-}3 \cdot 10^{-2} \mu m^3/s$. Interestingly, the intra-astrocytic and extracellular fluid velocities induced in our study result directly from the chemo-mechanical interactions following extracellular $K^+$ influx.

Part of our interest in interstitial fluid flow and the interplay between electrochemical activity and fluid dynamics relate to the intriguing effects of sleep on brain signaling, brain solute transport, and clearance [1, 59–61]. Both sleep spindles (11-16 Hz) and slow oscillations, i.e., low frequency (<1Hz) waves of neural activity, may support memory formation and learning [1, 62]. Fultz et al. [1] used multi-modal imaging to reveal coherent patterns of slow neural-, hemodynamical- and cerebrospinal fluid waves and suggest that these waves contribute to enhance brain clearance during sleep. Our results indicate that slow and ultraslow local waves of extracellular $K^+$ and $Na^+$ flux can induce pulsatile flow of interstitial fluid in the extracellular space as well as in astrocyte networks. The slow and ultraslow regimes express different electrochemical and fluid flow wave characteristics, including differences in wave amplitudes and phase shifts.

It is more challenging to compare the absolute and relative hydrostatic pressures obtained here with clinical or experimental measurements. The coupling between electro-chemical and mechanical effects leads to an inherent cascade in which ionic concentration differences induce an osmotic pressure difference across the membrane, transmembrane water flux and cellular swelling, and intracompartmental fluid flow. The difference $p_i - p_e$ between the hydrostatic pressures $p_i$ and $p_e$ is regulated by the intracellular volume changes resulting from transmembrane water movement, modulated by the elastic stiffness $K_m$ of the cell membrane. On

the other hand, the absolute value of these hydrostatic pressures are determined by the incompressibility of the fluid environment and the permeability of each compartment. Within this paradigm, our simulations suggest that the ion dynamics can induce localized differences in extracellular hydrostatic pressures of several kPa over a distance of 150 μm, which would correspond to an average spatial gradient of tens of MPa/m. These values are out of range when compared with e.g. the intracranial pressure (ICP), which pulsates with the cardiac and respiratory cycles with (supine) mean ICP values of $\sim$7–15 mmHg relative to baseline atmospheric pressure (1 mmHg = $\sim$133 Pa) in healthy subjects [63, 64], mean ICP wave amplitudes of $\sim$1.5–7 mmHg [65, 66], and spatial differences of less than 1–3 mmHg/m [67, 68]; but more comparable with normal cerebral perfusion pressure (representing the difference between the mean arterial pressure and the ICP) of 50—150 mmHg (6.7—20 kPa). In comparison, an osmotic pressure of 1 kPa (7.5 mmHg) across a cellular membrane corresponds to an osmotic concentration difference of only $\sim$0.4 mM. Future modeling work is required to couple e.g. vascular or perivascular pressure pulsations with the interstitial dynamics presented here.

Fluid flow may enhance ion and solute transport by advection in addition to the intrinsically present diffusion. We here discuss the advective contribution to $K^+$ transport through astrocytic networks. Previous computational studies of spatial $K^+$ buffering has utilized either models based on cable theory, where contributions from diffusive currents are neglected [69–72], or applied electrodiffusive frameworks accounting for the coupling of spatiotemporal variations in ion concentrations and electrical potentials [34, 73]. In contrast to the model presented here, neither of these previous models account for cellular swelling and advective transport. We find that $K^+$ is mainly transported through the ICS, and notably that electrical drift dominates both diffusive and advective transport. Interestingly, we observe a net transport of $K^+$ away from the input zone even in model scenario M1, where diffusion drives $K^+$ in the opposite direction (i.e., towards the input zone). The strong dominance of electrical drift in ICS $K^+$ transport is in accordance with the findings reported in Halnes et al. [34] and Zhu et al. [39]. Further, our findings indicate that osmotically driven flow through the astrocytic syncytium facilitates spatial buffering via advection: our model scenario M3 predicts a 21.5% higher $K^+$ transport (away from the input zone) than Halnes et al. [34].

Although the exact mechanisms for water transport across astrocytic membranes are debated [3], it is well established that astrocytes have higher water permeability than neurons in part since neurons do not express AQP4 [74]. The high astrocytic water permeability has been hypothesized to stabilize extracellular ion concentrations, shielding neurons from severe swelling [34]. Our findings indeed indicate that cellular swelling, and importantly osmotic transport within the astrocytic network, facilitate in preventing water potential build-up in the ECS and thus neuronal swelling. Further, we find that models for ionic electrodiffusion must account for fluid dynamics and cellular swelling to estimate osmotic pressure across glial-ECS membranes adequately.

The computational model considered here is complex, with numerous model parameters, giving rise to considerable uncertainty. Our sensitivity analysis shows that the maximum superficial fluid velocity varies substantially under variations in a selection of the parameters related to mechanics (compartmental permeability, membrane stiffness, and membrane water permeability). Still, the differences we observe between M1 and M3 are robust to the choice of these model parameters. As such, we deem the current model useful for pointing at a mechanistic understanding of how astrocytic response to neuronal activity may impact fluid movement in the brain.

In order to maintain a reasonable level of complexity, our model distinctly includes electrodiffusion, osmosis, and hydrostatic pressures, while the representation of a number of other mechanisms are substantially simplified. In particular, we let the transmembrane water

transport be parameterized by a single transmembrane water permeability ($\eta_\mathrm{m}$), assuming that fluid is carried by passive transporters only. The parameter $\eta_\mathrm{m}$ captures the permeability of all (passive) membrane fluid transporters lumped together. However, through which channels fluid flows and in what direction is still an unresolved issue [3] that could be studied further using the here proposed model as a starting point. Furthermore, we have not allowed fluid to enter or leave the system (i.e., closed boundary conditions). A natural next step for this modeling work would be to open up the boundaries to investigate the coupling between the brain tissue/neuropil and e.g. perivascular spaces. By allowing for fluid and ionic flow across the domain boundaries, one may model and explore the interplay between vascular or perivascular pressure pulsations at different frequencies and amplitudes, electrochemical activity, and interstitial fluid flow. In particular, this approach could be used to study the polarization of AQP4 channels, which are known to be densely packed at the astrocyte endfeet [75].

We conclude that the framework presented here is a promising tool for predicting complex phenomena related to electro-chemical-mechanical interplay in brain tissue. We point at a mechanistic understanding of how astrocytic response to neuronal activity and permeabilities may impact fluid movement in the brain. Our sensitivity analysis supports the idea that reduced glial water permeability may reduce (ICS and) ECS fluid velocities. Further *in-silico* studies with more physiological transmembrane water mechanisms and boundary conditions representing vascular or perivascular pressure pulsations could elucidate open questions related to the role of AQP4 and astrocytes in brain water clearance and homeostasis.

## 4 Methods

The homogenized tissue is represented by a one-dimensional domain $\Omega$, with length $L$ and outer boundary $\Gamma = \partial\Omega$. We assume that the tissue consists of two compartments representing the ICS (denoted by subscript $r = \mathrm{i}$) and the ECS (denoted by $r = \mathrm{e}$). We predict the evolution in time and distribution in space of the volume fractions $\alpha_r$, the ion concentrations $[\mathrm{Na}^+]_r$, $[\mathrm{K}^+]_r$, and $[\mathrm{Cl}^-]_r$, the electrical potentials $\phi_r$, and the hydrostatic pressures $p_r$. The model is embedded in the electrodiffusive Kirchhoff-Nernst-Planck framework and builds on previous work on ionic electrodiffusion [34, 38], fluid dynamics [39], and astrocyte modeling [34].

### 4.1 Governing equations

We consider the following system of coupled, time-dependent, nonlinear partial differential equations. Find the ICS volume fraction $\alpha_\mathrm{i} : \Omega \times (0, T] \to [0, 1]$ such that for each $t \in (0, T]$:

$$\frac{\partial\alpha_\mathrm{i}}{\partial t} + \nabla \cdot (\alpha_\mathrm{i} u_\mathrm{i}) = -\gamma_\mathrm{m} w_\mathrm{m}, \tag{1}$$

where $u_\mathrm{i} : \Omega \times (0, T] \to \mathbb{R}$ (m/s) is the ICS fluid velocity field. The transmembrane water flux $w_\mathrm{m}$ is driven by osmotic and hydrostatic pressure, and will be discussed further in Section 4.3. The coefficient $\gamma_\mathrm{m}$ (1/m) represents the area of cell membrane per unit volume of tissue. By definition, the total volume fractions sum to 1, and we assume that neurons occupy 40% of the total tissue volume [34, 76]. We thus have that:

$$\alpha_\mathrm{e} = (1 - 0.4) - \alpha_\mathrm{i}. \tag{2}$$

Further, for each ion species $k \in \{\mathrm{Na}^+, \mathrm{K}^+, \mathrm{Cl}^-\}$ and for $r = \{\mathrm{i}, \mathrm{e}\}$, find the ion concentration $[k]_r : \Omega \times (0, T] \to \mathbb{R}$ and the electrical potential $\phi_r : \Omega \times (0, T] \to \mathbb{R}$ such that for each $t \in$

$(0, T]$:

$$\frac{\partial(\alpha_i[k]_i)}{\partial t} + \nabla \cdot (\alpha_i j_i^k) = -\gamma_m j_m^k, \tag{3a}$$

$$\frac{\partial(\alpha_e[k]_e)}{\partial t} + \nabla \cdot (\alpha_e j_e^k) = \gamma_m (j_m^k + j_{input}^k + j_{decay}^k), \tag{3b}$$

where $j_i^k = j_i^k(x, t)$ and $j_e^k = j_e^k(x, t)$ (mol/(m$^2$s)) are the compartmental ionic flux densities for each ion species $k$. Modeling of the transmembrane ion flux density $j_m^k$, the input ion flux density $j_{input}^k$, and the decay ion flux density $j_{decay}^k$ will be discussed further in Sections 4.4–4.5. Note that (3) follows from first principles and express conservation of ion concentrations in each region. Moreover, we assume that the ion flux densities satisfy:

$$-\sum_k z_k \nabla \cdot (\alpha_i j_i^k) - \gamma_m \sum_k z_k j_m^k = 0, \tag{4a}$$

$$-\sum_k z_k \nabla \cdot (\alpha_e j_e^k) + \gamma_m \sum_k z_k j_m^k = 0, \tag{4b}$$

where $z_k$ (unitless) is the valence of ion species $k$. Note that (4) arises from assuming electroneutrality, i.e., that the sum of all charge inside a compartment is zero. Given the smallness of the capacitance and that we do not model action potentials, this is a well-established approximation to use in place of the charge-capacitor relation that is commonly used when modeling electrodiffusion [38]. We further assume that the compartmental ionic flux densities $j_r^k$ : $\Omega \times (0, T] \rightarrow \mathbb{R}$ are driven by diffusion, electric drift, and advection:

$$j_r^k = -\frac{D_k}{\lambda_r^2} \nabla[k]_r - \frac{D_k z_k}{\lambda_r^2 \psi} [k]_r \nabla \phi_r + u_r [k]_r, \quad r = i, e. \tag{5}$$

Here, $D_k$ (m$^2$/s) denotes the diffusion coefficient of ion species $k$ and $\lambda_r$ (unitless) denotes the tortuosity of compartment $r$. The constant $\psi = RT/F$ combines Faraday's constant $F$ (C/mol), the absolute temperature $T$ (K), and the gas constant $R$ (J/(molK)).

We now turn to the dynamics of the fluid velocity fields $u_r$ and the hydrostatic pressures $p_r : \Omega \times (0, T] \rightarrow \mathbb{R}$ (Pa). We will consider three different models (M1, M2, and M3) for the compartmental fluid velocities $u_i$ and $u_e$ that are detailed in Section 4.2. The relation between the intra- and extracellular hydrostatic pressure $p_i$ and $p_e$ is given by the force balance on the membrane [38, 39]:

$$p_i - p_e = K_m (\alpha_i - \alpha_{i,init}) + p_{m,init}, \tag{6}$$

here $K_m$ (Pa) denotes the membrane stiffness, $\alpha_{i,init}$ (unitless) denotes the initial intracellular volume fraction, and $p_{m,init}$ (Pa) is the initial hydrostatic pressure difference across the membrane (see Table 2). Furthermore, we assume that the volume–fraction weighted fluid velocity is divergence free; that is:

$$\nabla \cdot \left( \sum_r \alpha_r u_r \right) = 0. \tag{7}$$

By inserting (6) and the relevant expressions for the compartmental fluid velocities $u_i$ and $u_e$ (see Section 4.2 for further details) into (7), we obtain an equation for the extracellular hydrostatic pressure.

**Table 2. Model parameters.**

| Symbol | Definition | Value | Ref. |
|---|---|---|---|
| $L$ | Length of domain | $3.0 \cdot 10^{-4}$ m | [34] |
| $F$ | Faraday's constant | 96 485.3 C/mol | |
| $R$ | Gas constant | 8.314 J/(mol K) | |
| $T$ | Temperature | 310.15 K | |
| $i$ | Van't Hoff factor | 1 | |
| $D_{Na}$ | Na$^+$ diffusion constant | $1.33 \cdot 10^{-9}$ m$^2$/s | [34] |
| $D_{K}$ | K$^+$ diffusion constant | $1.96 \cdot 10^{-P}$ m$^2$/s | [34] |
| $D_{Cl}$ | Cl$^-$ diffusion constant | $2.03 \cdot 10^{-9}$ m$^2$/s | [34] |
| $\lambda_i$ | Intracellular tortuosity | 3.2 | [34] |
| $\lambda_e$ | Extracellular tortuosity | 1.6 | [34] |
| $\gamma_m$ | Membrane area per unit volume of tissue | $8 \cdot 10^{+6}$ 1/m | [34] |
| $\bar{g}_{Na}$ | Membrane conductance for Na$^+$ | 1 S/m$^2$ | [34] |
| $\bar{g}_{K}$ | Membrane conductance for K$^+$ | 16.96 S/m$^2$ | [34] |
| $\bar{g}_{Cl}$ | Membrane conductance for Cl$^-$ | 0.5 S/m$^2$ | [34] |
| $\rho_{pump}$ | Maximum pump rate | $1.12 \cdot 10^{-6}$ mol/(m$^2$s) | [34] |
| $P_{Nai}$ | [Na$^+$]$_i$ threshold for Na$^+$/K$^+$ pump | 10 mol/m$^3$ | [34] |
| $P_{Ke}$ | [K$^+$]$_e$ threshold for Na$^+$/K$^+$ pump | 1.5 mol/m$^3$ | [34] |
| $\eta_m$ | Membrane water permeability | $8.14 \cdot 10^{-14}$ m/(Pa s) | [35] |
| $K_m$ | Membrane stiffness | $2.294 \cdot 10^3$ Pa | [77] |
| $\kappa_i$ | ICS permeability | $1.8375 \cdot 10^{-14}$ m$^2$/(Pa s) | |
| $\kappa_e$ | ECS permeability | $1.8375 \cdot 10^{-14}$ m$^2$/(Pa s) | [27] |
| $\epsilon_r$ | Relative permittivity of the ECS | 84.6 | [41] |
| $\epsilon_0$ | Vacuum permittivity | $8.85 \cdot 10^{-12}$ F/m | [41] |
| $\zeta$ | Zeta-potential | $-22.8 \cdot 10^{-3}$ V | [41] |
| $\mu$ | Viscosity of water | $6.4 \cdot 10^{-4}$ Pas | [41] |
| $p_{m,init}$ | Initial transmembrane hydrostatic pressure difference | $1 \cdot 10^3$ Pa | [78] |
| $j_{in}^{M0}$ | Constant input flux density (M0) | $8.28 \cdot 10^{-7}$ mol/(m$^2$s) | |
| $j_{in}^{M1}$ | Constant input flux density (M1) | $8.0 \cdot 10^{-7}$ mol/(m$^2$s) | |
| $j_{in}^{M2}$ | Constant input flux density (M2) | $9.15 \cdot 10^{-7}$ mol/(m$^2$s) | |
| $j_{in}^{M3}$ | Constant input flux density (M3) | $9.05 \cdot 10^{-7}$ mol/(m$^2$s) | |
| $k_{dec}$ | Decay factor for [K$^+$]$_e$ | $2.9 \cdot 10^{-8}$ m/s | [34] |

The combination of (1), (3), (4), and (7) with insertion of (5) and (6), and the expressions for $u_r$ (c.f. Section 4.2) define a system of 10 differential equations for the 10 unknown fields ($\alpha_i$, $[k]_r$, $\phi_r$, and $p_e$). Note that the extracellular volume fraction $\alpha_e$ and the intracellular hydrostatic pressure $p_i$ can be calculated using respectively (2) and (6). Appropriate initial conditions, boundary conditions, and importantly membrane mechanisms close the system.

## 4.2 Expressions for fluid velocities (model scenarios M1, M2, and M3)

To model compartmental fluid flow, we consider three different modeling setups:

**M1** We assume that the compartmental fluid flow is driven by hydrostatic pressure gradients. Specifically, the fluid velocities are given by:

$$u_i = -\kappa_i \nabla p_i, \tag{8a}$$

$$u_e = -\kappa_e \nabla p_e, \tag{8b}$$

where $\kappa_r$ (m$^2$/(Pa s)) denotes the mobility of compartment $r$. We will refer to $\kappa_r$ as the compartmental permeability.

**M2** We assume that fluid flow in the glial network is driven by hydrostatic and osmotic pressure gradients. Since Na$^+$, K$^+$, and Cl$^-$ can move through the ICS via gap junctions, we assume that they do not contribute to osmosis. Thus, only the immobile ions drive the osmotic flow. As osmotic forces only act across membranes, we assume that fluid flow in the ECS is only driven by hydrostatic pressure gradients. The fluid velocities are given by:

$$u_i = -\kappa_i(\nabla p_i - iRT \nabla \frac{a_i}{\alpha_i}), \tag{9a}$$

$$u_e = -\kappa_e \nabla p_e, \tag{9b}$$

where $i$ (unitless) is the Van't Hoff factor, and $a_i$ (mol/m$^3$) is the concentration of immobile ions.

**M3** We follow Zhu et al. 2021 [39] and assume that the intracellular fluid flow is driven by hydrostatic and osmotic forces, while the extracellular fluid flow is driven by hydrostatic and electro-osmotic forces. To model the electro-osmotic flow, we use the Helmholtz-Smoluchowski approximation [41]. The fluid velocities are given by:

$$u_i = -\kappa_i(\nabla p_i - iRT \nabla \frac{a_i}{\alpha_i}), \tag{10a}$$

$$u_e = -\kappa_e \nabla p_e - \frac{\epsilon_r \epsilon_0 \zeta}{\mu} \nabla \phi_e. \tag{10b}$$

Here, $\epsilon_r$ (unitless) is the relative permittivity of the extracellular solution, $\epsilon_0$ (F/m) is the vacuum permittivity, $\zeta$ (V) is the zeta-potential, and $\mu$ (Pas) is the viscosity of water.

## 4.3 Transmembrane fluid flow

The transmembrane fluid flow is driven by hydrostatic and osmotic pressure gradients, and the fluid velocity, $w_m$ (m/s), is expressed as:

$$w_m = \eta_m(p_i - p_e + iRT(O_e - O_i)). \tag{11}$$

Here, $\eta_m$ (m/(Pas)) is the membrane water permeability and $O_r$ is the osmolarity of compartment $r$. We assume that all ion species contribute to the osmolarity, which is given by

$$O_r = \frac{a_r}{\alpha_r} + \sum_k [k]_r. \tag{12}$$

Multiplying $O_r$ by $-iRT$ gives us the solute potential in compartment $r$, $\Pi_r$.

## 4.4 Membrane mechanisms

We adopt the ionic membrane mechanisms from Halnes et al. 2013 [34]. The mechanisms include a Na$^+$ leak channel, a Cl$^-$ leak channel, an inward-rectifying K$^+$ channel, and a Na$^+$/K$^+$ pump. The membrane flux densities (mol/(m$^2$s)) are given by:

$$j_m^{Na} = \frac{\bar{g}_{Na}}{Fz_{Na}}(\phi_m - E_{Na}) + 3j_{pump}, \tag{13a}$$

$$j_{\mathrm{m}}^{\mathrm{K}} = \frac{\bar{g}_{\mathrm{K}} f_{\mathrm{Kir}}}{F z_{\mathrm{K}}} (\phi_{\mathrm{m}} - E_{\mathrm{K}}) - 2 j_{\mathrm{pump}}, \tag{13b}$$

$$j_{\mathrm{m}}^{\mathrm{Cl}} = \frac{\bar{g}_{\mathrm{Cl}}}{F z_{\mathrm{Cl}}} (\phi_{\mathrm{m}} - E_{\mathrm{Cl}}), \tag{13c}$$

where $\bar{g}_k$ (S/m$^2$) is the membrane conductance for ion species $k$, $\phi_{\mathrm{m}}$ (V) is the membrane potential (defined as $\phi_{\mathrm{i}} - \phi_{\mathrm{e}}$), and $E_k$ (V) is the reversal potential. The reversal potentials are given by the Nernst equation:

$$E_k = \frac{RT}{F z_k} \ln \left( \frac{[k]_{\mathrm{e}}}{[k]_{\mathrm{i}}} \right). \tag{14}$$

Further, the Kir-function, $f_{\mathrm{Kir}}$, which modifies the inward-rectifying K$^+$ current, is given by:

$$f_{\mathrm{Kir}}([\mathrm{K}^+]_{\mathrm{e}}, \Delta\phi, \phi_{\mathrm{m}}) = \sqrt{\frac{[\mathrm{K}^+]_{\mathrm{e}}}{[\mathrm{K}^+]_{\mathrm{e,init}}}} \left[ \frac{AB}{CD} \right],$$

where

$$A = 1 + \exp(18.4/42.4), \qquad B = 1 + \exp(-(0.1186 + E_{\mathrm{K,init}})/0.0441),$$
$$C = 1 + \exp((\Delta\phi + 0.0185)/0.0425), \quad D = 1 + \exp(-(0.1186 + \phi_{\mathrm{m}})/0.0441).$$

Here, $\Delta\phi = \phi_{\mathrm{m}} - E_k$, and $E_{\mathrm{K,init}}$ is the reversal potential for K$^+$ at initial ion concentrations. Finally, the pump flux density (mol/(m$^2$s)) is given by:

$$j_{\mathrm{pump}} = \rho_{\mathrm{pump}} \left( \frac{[\mathrm{Na}^+]_{\mathrm{i}}^{1.5}}{[\mathrm{Na}^+]_{\mathrm{i}}^{1.5} + P_{\mathrm{Nai}}^{1.5}} \right) \left( \frac{[\mathrm{K}^+]_{\mathrm{e}}}{[\mathrm{K}^+]_{\mathrm{e}} + P_{\mathrm{Ke}}} \right), \tag{15}$$

where $\rho_{\mathrm{pump}}$ (mol/(m$^2$s)) is the maximum pump rate, $P_{\mathrm{Nai}}$ (mol/m$^3$) is the $[\mathrm{Na}^+]_{\mathrm{i}}$ threshold, and $P_{\mathrm{Ke}}$ (mol/m$^3$) is the $[\mathrm{K}^+]_{\mathrm{e}}$ threshold.

## 4.5 Input/decay fluxes

To stimulate the system, we follow the same procedure as in Halnes et al. 2013 [34]. We assume that there is a group of highly active neurons within the input zone, defined to be the interval $[L_1, L_2]$ with $L_1 = 1.35 \cdot 10^{-4}$ m and $L_2 = 1.65 \cdot 10^{-4}$ m, during the time interval [10 s, 210 s]. The neurons are not modeled explicitly, but we mimic their activity by injecting a K$^+$ current into the ECS and removing the same amount of Na$^+$ ions simultaneously. The input flux $j_{\mathrm{input}}^k : [L_1, L_2] \times [10\text{ s}, 210\text{ s}] \to \mathbb{R}$ is given by three different protocols:

$$j_{\mathrm{input}}^{\mathrm{K}} = -j_{\mathrm{input}}^{\mathrm{Na}} = \begin{cases} j_{\mathrm{in}}, & \text{constant stimulus} \\ (j_{\mathrm{in}}/2)\sin(2\pi t) + j_{\mathrm{in}}/2, & \text{slow stimulus} \\ (j_{\mathrm{in}}/2)\sin(2\pi t 0.05) + j_{\mathrm{in}}/2, & \text{ultraslow stimulus} \end{cases} \tag{16}$$

where $j_{\mathrm{in}}$ (mol/(m$^2$s)) is constant. The constant stimulus is used for all simulations shown throughout this paper, except for the simulations illustrated by dark green and blue lines in Figs 7 and 8, which are run using the slow (1 Hz) and ultraslow (0.05 Hz) stimuli.

To mimic neuronal pumps and cotransporters, we remove K$^+$ ions from the extracellular space at a given decay rate and add the same amount of Na$^+$ ions. The decay is proportional to the extracellular K$^+$ concentration and defined across the whole domain. Specifically, the decay current $j_{\text{decay}}^k : \Omega \times (0, T] \to \mathbb{R}$ (mol/(m$^2$s)) is given by:

$$j_{\text{decay}}^K = -j_{\text{decay}}^{Na} = -k_{\text{dec}}([K^+]_e - [K^+]_{e,\text{init}}),\qquad(17)$$

where $k_{\text{dec}}$ denotes the $[K^+]_e$ decay factor, and $[K^+]_{e,\text{init}}$ the initial extracellular K$^+$ concentration.

## 4.6 Model parameters

All parameters of the system are as listed in Table 2.

**4.6.1 Choice of membrane stiffness.** Lu et al. 2006 [77, Fig 1D, Fig S6] report the Young's modulus $E$ of individual astrocytes in the hippocampus for different deformation frequencies with $E \approx 300$ Pa for 30 Hz, $E \approx 420$ Pa for 100 Hz, and $E \approx 520$ Pa for 200 Hz, and a Poisson's ratio $v = 0.47$. We use the mean Young's modulus (413 Pa) to calculate the bulk modulus (stiffness constant) $K_m$ as:

$$K_m = \frac{E}{3(1 - 2v)} = 2.294 \cdot 10^3 \text{Pa}.$$

## 4.7 Boundary conditions

We apply sealed-end boundary conditions to the system, that is, no ions and no fluid are allowed to enter or leave the system on the boundary $\Gamma$:

$$\alpha_r j_r^k \cdot n_\Gamma \quad = 0 \quad \text{on } \Gamma,\qquad(18a)$$

$$\alpha_r u_r \cdot n_\Gamma \quad = 0 \quad \text{on } \Gamma,\qquad(18b)$$

where $n_\Gamma$ is the outward pointing normal vector.

The extracellular electrical potential, $\phi_e$, and the extracellular hydrostatic pressure, $p_e$, are only determined up to constants. To constrain the electrical potential, we require that:

$$\int_\Omega \phi_e dx = 0.\qquad(19)$$

We enforce this zero-average constraint by introducing an additional unknown (a Lagrange multiplier) $c_e$. For the extracellular hydrostatic pressure, we set

$$p_e = 0 \quad \text{on } \Gamma_{\text{right}}.\qquad(20)$$

## 4.8 Initial conditions

We obtain initial conditions for the system through a two-step procedure. First, we specify a set of pre-calibrated initial values (Table 3, Pre-calibrated column). Second, we calibrate the model by running a simulation for $1 \cdot 10^6$ s. For the calibration, we set the transmembrane water permeability to zero and use $N = 100$ and $\Delta t = 10^{-2}$. The final values from the calibration are listed in Table 3 (Post-calibrated column).

**Table 3. Initial conditions and baseline values.**

| Variable | Pre-calibrated | Post-calibrated* | Ref. |
|---|---|---|---|
| $\alpha_i$ | 0.4 | 0.4 | [34] |
| $[Na^+]_i$ | 15.189 mM | 15.475 mM | [34] |
| $[Na^+]_e$ | 144.662 mM | 144.091 mM | [34] |
| $[K^+]_i$ | 99.959 mM | 99.892 mM | [34] |
| $[K^+]_e$ | 3.082 mM | 3.216 mM | [34] |
| $[Cl^+]_i$ | 5.145 mM | 5.364 mM | [34] |
| $[Cl^-]_e$ | 133.71 mM | 133.273 mM | [34] |
| $\alpha_e$ | | 0.2 | Eq 2 |
| $\phi_m$ | | −85.9 mV | Eq 4 |
| $p_i$ | | $1 \cdot 10^3$ Pa | Eq 6 |
| $p_e$ | | 0.0 Pa | Eq 7 |

* Values with more significant digits included were used in the simulations. (Available with the source code.)

To ensure fluid equilibrium at $t = 0$ s and electroneutrality of the system, we define a set of immobile macromolecules, $a_i$ and $a_e$ (mol/m$^3$), with charge number $z_0$ (unitless) based on the initial ion concentrations. These are defined as constant concentrations with respect to the total volume of the system. Requiring an electroneutral system and zero transmembrane fluid flux at $t = 0$ s gives:

$$F\left(\sum_k z_k[k]_{e,init} + z_0 \frac{a_e}{\alpha_{e,init}}\right) = 0, \tag{21a}$$

$$F\left(\sum_k z_k[k]_{i,init} + z_0 \frac{a_i}{\alpha_{i,init}}\right) = 0, \tag{21b}$$

$$\eta_m\left(p_{m,init} + RT\left(\frac{a_e}{\alpha_{e,init}} + \sum_k [k]_{e,init} - \frac{a_i}{\alpha_{i,init}} - \sum_k [k]_{i,init}\right)\right) = 0. \tag{21c}$$

Solving (21) gives the following expressions for $z_0$, $a_i$, and $a_e$:

$$z_0 = \frac{\sum_k z_k[k]_{e,init} - \sum_k z_k[k]_{i,init}}{\frac{p_{m,init}}{RT} + \sum_k [k]_{e,init} - \sum_k [k]_{i,init}}, \tag{22a}$$

$$a_i = -\sum_k z_k[k]_{i,init} \frac{\alpha_{i,init}}{z_0}, \tag{22b}$$

$$a_e = -\sum_k z_k[k]_{e,init} \frac{\alpha_{e,init}}{z_0}. \tag{22c}$$

To ensure strict electroneutrality and fluid equilibrium, we calculate the values at the beginning of each simulation. By using the post-calibrated initial conditions listed in Table 3, the values are approx. $z_0 = -0.6$, $a_e = 4.7$ mM, and $a_i = 73.5$ mM. Note that $z_0$ is interpreted as the

average charge number of the macromolecules, and may thus be a decimal number (e.g., if not all the immobile macromolecules added to the system are charged).

### 4.9 Numerical implementation and verification

We discretize the system using a finite element method in space with characteristic mesh size $\Delta x$ and a first order implicit finite difference scheme in time with time step $\Delta t$. Our numerical scheme and implementation builds on previous work presented in Ellingsrud et al. 2021 [40].

The numerical scheme is implemented via the FEniCS finite element library [79] (Python 3.8), and the code is openly available at https://github.com/martejulie/fluid-flow-in-astrocyte-networks. To determine the time step $\Delta t$ and the mesh size $\Delta x$, we perform a numerical convergence study. Specifically, we apply the numerical scheme outlined above for model scenario M1 for different mesh resolutions and time steps: $\Delta x = L/N$ for $N = 25, 50, 100, 200, 400, 800$ and $\Delta t = 1, 10^{-1}, 10^{-2}, 10^{-3}, 10^{-4}$ s. For each simulation, we calculate the peak extracellular superficial fluid velocity and the peak concentration of ECS $K^+$ at $t = 20$ s. We find that the peak extracellular superficial fluid velocity and the ECS $K^+$ concentration converge towards 0.271 $\mu$m/min and 9.186 mM, respectively, as the temporal and spatial resolution increase (Table 4). Based on our findings, we choose $\Delta t = 10^{-3}$s and $N = 400$ for all simulations presented within this work.

### 4.10 Calculation of advection/diffusion and advection/drift fractions

We calculate the advection/diffusion fraction $F_{\text{diff}}$ and the advection/electric-drift fraction $F_{\text{drift}}$ for ion species $k$ as follows:

$$F_{\text{diff}}^k = |j_{\text{adv}}^k/j_{\text{diff}}^k|, \tag{23}$$

$$F_{\text{drift}}^k = |j_{\text{adv}}^k/j_{\text{drift}}^k|, \tag{24}$$

where $j_{\text{adv}}^k$ (mol/(m$^2$s)), $j_{\text{diff}}^k$ (mol/(m$^2$s)), and $j_{\text{drift}}^k$ (mol/(m$^2$s)) are the advective, diffusive, and electric-drift components of the peak total ionic flux at $t = 200$ s, respectively.

**Table 4. Numerical verification.** Quantities of interest converge under spatial and temporal discrete refinement.

| $\Delta t$ \ $N$ | 25 | 50 | 100 | 200 | 400 | 800 |
|---|---|---|---|---|---|---|
| 1 | 0.217 | 0.268 | 0.269 | 0.271 | 0.272 | 0.272 |
| $10^{-1}$ | 0.216 | 0.267 | 0.268 | 0.270 | 0.270 | 0.271 |
| $10^{-2}$ | 0.216 | 0.267 | 0.268 | 0.271 | 0.271 | 0.271 |
| $10^{-3}$ | 0.216 | 0.267 | 0.268 | 0.271 | 0.271 | 0.271 |
| $10^{-4}$ | 0.216 | 0.267 | 0.268 | 0.271 | 0.271 | 0.271 |

**(a)** $\max(\alpha_e u_e)$ ($\mu$m/ min)

| $\Delta t$ \ $N$ | 25 | 50 | 100 | 200 | 400 | 800 |
|---|---|---|---|---|---|---|
| 1 | 7.799 | 9.204 | 9.198 | 9.205 | 9.208 | 9.207 |
| $10^{-1}$ | 7.774 | 9.174 | 9.169 | 9.175 | 9.177 | 9.177 |
| $10^{-2}$ | 7.780 | 9.182 | 9.176 | 9.183 | 9.185 | 9.185 |
| $10^{-3}$ | 7.781 | 9.183 | 9.177 | 9.184 | 9.186 | 9.186 |
| $10^{-4}$ | 7.781 | 9.183 | 9.177 | 9.184 | 9.186 | 9.186 |

**(b)** $\max([K^+]_e)$ (mM)

## Acknowledgments

We want to thank Yoichiro Mori for valuable discussions and Jørgen S. Dokken for help with setting up the GitHub repositories.

## Author Contributions

**Conceptualization:** Marte J. Sætra, Ada J. Ellingsrud, Marie E. Rognes.

**Formal analysis:** Marte J. Sætra, Ada J. Ellingsrud.

**Funding acquisition:** Marie E. Rognes.

**Investigation:** Marte J. Sætra, Ada J. Ellingsrud.

**Methodology:** Marte J. Sætra, Ada J. Ellingsrud, Marie E. Rognes.

**Project administration:** Marie E. Rognes.

**Software:** Marte J. Sætra, Ada J. Ellingsrud.

**Supervision:** Marie E. Rognes.

**Validation:** Marte J. Sætra, Ada J. Ellingsrud.

**Visualization:** Marte J. Sætra, Ada J. Ellingsrud, Marie E. Rognes.

**Writing – original draft:** Marte J. Sætra, Ada J. Ellingsrud, Marie E. Rognes.

**Writing – review & editing:** Marte J. Sætra, Ada J. Ellingsrud, Marie E. Rognes.

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
