## [Decision Letter · Decision Letter 0]

27 Mar 2023

Dear Dr. Sætra,

Thank you very much for submitting your manuscript "Neural activity induces strongly coupled electro-chemo-mechanical interactions and fluid flow in astrocyte networks and extracellular space - a computational study" for consideration at PLOS Computational Biology.

As with all papers reviewed by the journal, your manuscript was reviewed by members of the editorial board and by several independent reviewers. In light of the reviews (below this email), we would like to invite the resubmission of a significantly-revised version that takes into account the reviewers' comments.  Both reviewers appreciated the importance and significance of the work, and both would like to see additional simulations in a revised manuscript.

We cannot make any decision about publication until we have seen the revised manuscript and your response to the reviewers' comments. Your revised manuscript is also likely to be sent to reviewers for further evaluation.

Sincerely,

Kim T. Blackwell, V.M.D., Ph.D.

Academic Editor

PLOS Computational Biology

Thomas Serre

Section Editor

PLOS Computational Biology

Reviewer's Responses to Questions

**Comments to the Authors:**

Reviewer #1: This manuscript presents a numerical study of fluid flow, ion motion, and associated pressures and electrical potentials, in and around an astrocyte in the brain. The work is motivated by recent interest in motion of interstitial fluid and cerebrospinal fluid, which has implications for edema, stroke, vascular dementia, and neurodegenerative disease. The ions considered include K+, Na+, and Cl-. The astrocyte wall is modeled as elastic, allowing cellular expansion and contraction. A set of ten coupled partial differential equations describes the system and is solved numerically in a time-dependent, one-dimensional domain. Neuronal activity is simulated by injecting K+ and Na+ near the center of the domain. A grid convergence study is presented. Four scenarios are studied: one in which fluid velocity is assumed to be zero (dubbed M0), one in which fluid velocity is driven by hydrostatic pressure gradients alone (M1), one in which hydrostatic pressure effects are retained and are complemented inside astrocytes by osmotic pressure gradients (M2), and one in which both of those effects are retained and are complemented outside astrocytes by electro-osmosis (M3). The resulting concentrations increase and decrease with curves of the sort we might expect (exponential, or nearly) and have values of the same order of magnitude as found in prior experiments and simulations. In scenario M1, the authors find a two-roll fluid circulation into astrocytes at the location of ion injection, along cell bodies, and out again at the domain edges. They also find that osmotic pressure differences between astrocytes and extracellular spaces are small in scenario M2 but larger, and generally similar, in scenarios M1 and M3. Fluid velocities are similar in scenarios M2 and M3. A sensitivity analysis shows that varying the compartmental permeability, membrane stiffness, and membrane permeability over reasonable ranges alters maximum transport rates relatively little.

The modeling and manuscript are thorough and meticulous. The results are clear enough and extend significantly beyond prior studies, which have not accounted for coupling among so many mechanisms. The interpretations are straightforward and not overwrought. Thus I strongly support publication, providing the authors respond effectively to a few questions and comments:

Figure 2 shows temporal variations that look like smooth exponential rise and decay in every case except one, the concentration of K+ ions (panel D). What is happening there?

The results consistently show that osmotic pressures are much higher than hydrostatic pressures. How much error would be introduced if hydrostatic pressures were neglected altogether?

The scenarios simulated all consider quasi-steady behaviors between step changes in neuronal activity (turning on at 10 s, then turning off at 210 s). Of course, many other timescales are of interest neurologically. Most of all I wonder what the model would predict for slow / ultraslow frequencies of the sort considered by Fultz et al. (Ref. 1). Could the authors run the model with neuronal activity varying at 0.05 Hz or 1 Hz? Or at least speculate in the Discussion about what might occur?

Finally I noticed (just two) typos:

line 387 - acknowledges should be acknowledge

line 153 - stabilize should be stabilizing

Reviewer #2: Sætra ploscompbio2023

Interest in the nature and drivers of fluid flow within the brain has taken on renewed interest with the idea that this movement may be important for clearance of waste from the brain through the glymphatic system. Because these flows are technically difficult to measure, they have been the focus of many computational models, though most of them have focused on mechanical drivers (pulsation, etc.). This paper investigates the intra and extracellular fluid flows induced by ionic/osmotic currents (driven by local neural activity) in simple model of the astrocyte syncytia and extracellular space. The authors find that there is fluid flow away from the site of activity and into the astrocytes remote from the site. The authors find that including osmotic and electrodiffusion in their models have a large effect on flow. There has been very little work done in this area and this looks to be an important contribution.

Major comments:

The 6.7mM K+ external change that is used here is very high, well beyond the standard physiological range. Increases in vivo are more in the range of 1 mM (see (Rasmussen, 2019) for some measurements on naturalistic conditions). That being said, looking at the equations I do not see anything that would prevent the flows scaling linearly (or something like that) with linear increases in injected /withdrawn K+/Na+. Can the authors make address this point, either analytically or with simulations if this is the case?

Will multiple areas of activity lead to a linear superposition of the observed dynamics?

The abstract and summary are vague and do not contain concise descriptions of the results. They should be re-written so that what is shown in the paper is summarized in the abstract.

I think the paper would be strengthened by exploring a different set of boundary conditions. The no-flow conditions here are fine as a simplification, but in the real system one might expect a constant pressure/ionic concentration boundary that might produce different responses, particularly if the activity location is moving. The “holy grail” of all this glymphatic system modeling is to get directed fluid flow (left to right or vice versa in this model’s extracellular space). Is there any way the authors could generate something like this, or suggest experiments that could test their ideas?

Minor comments:

Don’t use dotted lines- it makes it hard to see sharp transitions.

A zoom in on the onset offset dynamics in figure 2 would be helpful for the reader to understand the timescale of the changes here.

Mark default values on figure 7.

References

Rasmussen R, Nicholas E, Petersen NC, Dietz AG, Xu Q, Sun Q, Nedergaard M (2019) Cortex-wide Changes in Extracellular Potassium Ions Parallel Brain State Transitions in Awake Behaving Mice. Cell Rep 28:1182-1194 e1184. doi:10.1016/j.celrep.2019.06.082

**Have the authors made all data and (if applicable) computational code underlying the findings in their manuscript fully available?**

Reviewer #1: **No: **

Reviewer #2: Yes

PLOS authors have the option to publish the peer review history of their article (what does this mean?). If published, this will include your full peer review and any attached files.

Reviewer #1: **Yes: **Douglas Kelley

Reviewer #2: No
---

## [Decision Letter · Decision Letter 1]

28 Jun 2023

Dear Dr. Sætra,

We are pleased to inform you that your manuscript 'Neural activity induces strongly coupled electro-chemo-mechanical interactions and fluid flow in astrocyte networks and extracellular space - a computational study' has been provisionally accepted for publication in PLOS Computational Biology.

Best regards,

Kim T. Blackwell, V.M.D., Ph.D.

Academic Editor

PLOS Computational Biology

Thomas Serre

Section Editor

PLOS Computational Biology

Reviewer's Responses to Questions

**Comments to the Authors:**

Reviewer #1: The authors have addressed my queries expertly. I encourage publication.

Reviewer #2: The authors have addressed all my concerns.

**Have the authors made all data and (if applicable) computational code underlying the findings in their manuscript fully available?**

Reviewer #1: Yes

Reviewer #2: Yes

PLOS authors have the option to publish the peer review history of their article (what does this mean?). If published, this will include your full peer review and any attached files.

Reviewer #1: No

Reviewer #2: No

---

## [Editor Report · Acceptance letter]

13 Jul 2023

PCOMPBIOL-D-23-00344R1 

Neural activity induces strongly coupled electro-chemo-mechanical interactions and fluid flow in astrocyte networks and extracellular space - a computational study

Dear Dr Sætra,

I am pleased to inform you that your manuscript has been formally accepted for publication in PLOS Computational Biology. Your manuscript is now with our production department and you will be notified of the publication date in due course.

With kind regards,

Zsofia Freund
